# Aortic pressure and forward and backward wave components in children, adolescents and young-adults: Agreement between brachial oscillometry, radial and carotid tonometry data and analysis of factors associated with their differences

**Agustina Zinoveev**[1☯], **Juan M. Castro**[1], **Victoria García-Espinosa**[1], **Mariana Marin**[1], **Pedro Chiesa**[2], **Daniel Bia**[1☯], **Yanina Zócalo**[1☯]*

**1** Departamento de Fisiología, Facultad de Medicina, Centro Universitario de Investigación, Innovación y Diagnóstico Arterial (CUiiDARTE), Universidad de la República, Montevideo, Uruguay, **2** Servicio de Cardiología Pediátrica, Centro Hospitalario Pereira-Rossell, ASSE - Facultad de Medicina, Universidad de la República, Montevideo, Uruguay

☯ These authors contributed equally to this work.
* yana@fmed.edu.uy

## Abstract

Non-invasive devices used to estimate central (aortic) systolic pressure (cSBP), pulse pressure (cPP) and forward (Pf) and backward (Pb) wave components from blood pressure (BP) or surrogate signals differ in arteries studied, techniques, data-analysis algorithms and/or calibration schemes (e.g. calibrating to calculated [MBPc] or measured [MBPosc] mean pressure). The aims were to analyze, in children, adolescents and young-adults (1) the agreement between cSBP, cPP, Pf and Pb obtained using carotid (CT) and radial tonometry (RT) and brachial-oscillometry (BOSC); and (2) explanatory factors for the differences between approaches-data and between MBPosc and MBPc.1685 subjects (mean/range age: 14/3-35 y.o.) assigned to three age-related groups (3–12; 12–18; 18–35 y.o.) were included. cSBP, cPP, Pf and Pb were assessed with BOSC (Mobil-O-Graph), CT and RT (SphygmoCor) records. Two calibration schemes were considered: MBPc and MBPosc for calibrations to similar BP levels. Correlation, Bland-Altman tests and multiple regression models were applied. Systematic and proportional errors were observed; errors´ statistical significance and values varied depending on the parameter analyzed, methods compared and group considered. The explanatory factors for the differences between data obtained from the different approaches varied depending on the methods compared. The highest cSBP and cPP were obtained from CT; the lowest from RT. Independently of the technique, parameter or age-group, higher values were obtained calibrating to MBPosc. Age, sex, heart rate, diastolic BP, body weight or height were explanatory factors for the differences in cSBP, cPP, Pf or Pb. Brachial BP levels were explanatory factors for the differences between MBPosc and MBPc.

**Data Availability Statement:** The data underlying the results presented in the study are available from CUiiDARTE Centre (cuiidarte@fmed.edu.uy; Dr. Yanina Zócalo; http://www.fmed.edu.uy/).

**Funding:** This work was funded by Agencia Nacional de Investigación e Innovación (ANII), grant number: FSPI_X_2015_1_108484 - ANII (https://www.anii.org.uy/) to DB, YZ. The funders had no role in study design, data collection and analysis, decision to publish, or preparation of the manuscript.

**Competing interests:** The authors have declared that no competing interests exist.

## Introduction

The independent prognostic value of central aortic blood pressure (cBP) and aortic wave-derived parameters has been demonstrated[1–6]. That, together with the growing interest towards improving risk estimates contributes to explain the explosive development in the last decades, of methods and devices aiming at providing cBP, aortic wave components (e.g. forward and backward aortic pressure wave amplitude, Pf and Pb respectively) and/or derived parameters levels[3–7]. Available non-invasive devices estimate central systolic BP (cSBP) from pressure or surrogate signals obtained from peripheral arteries (e.g. carotid, brachial or radial), recorded using a variety of techniques (e.g. applanation tonometry, brachial oscillometry). From the obtained signals, and after calibrating them, the devices quantify cSBP directly (e.g. direct calibration of carotid waves) and/or indirectly, applying generalized transfer functions (GTF), low-pass filters or wave analysis[8,9]. Due to the differences in sites of measurement, signals recorded and in the methodological approaches used to assess central hemodynamics, data obtained from different approaches could differ and inter-device agreement would vary, depending on the parameter considered[8,9]. Although these approaches are already used in research involving children and adolescents [10–12], the extent to which they provide similar data on cBP and/or wave-derived parameters is unknown.

It is recognized that the pressure waveform is modified and the aortic or central pulse pressure (cPP) is amplified towards the periphery[13]. The centre-periphery changes in pressure wave are associated with age. The differences between cSBP and peripheral systolic blood pressure (pSBP)are greater in young subjects than in old adults, and may be particularly important in children and adolescents. Then, the relationship between central and peripheral hemodynamic parameters would vary depending on subjects´ age. This could affect the accuracy of estimating central parameters from peripheral data obtained with a methodological approach and/or the agreement between devices or methods. Hence, it would be valuable to analyze the agreement of methods used to estimate cBP and wave derived parameters considering data from subjects of different ages. It would be interesting to identify other subjects´ characteristics (e.g. demographic, anthropometric) that could contribute to explain the degree of agreement between data from different approaches.

An additional relevant issue to evaluate is the extent to which wave-derived parameters and/or cBP values depend on the calibration scheme considered: (1) calibrating to mean blood pressure (MBP) measured by oscillometry (MBPosc) or (2) calibrating to MBP calculated (MBPc), obtained from pSBP and peripheral diastolic blood pressure (pDBP). To know this, as well as to clarify which variables may influence the difference between MBPc and MBPosc would be valuable at the time of assessing and analyzing accurately cBP levels and/or wave-derived parameters.

This work aims were: 1) to determine the agreement of cSBP, cPP, Pf and Pb data obtained in children, adolescents and young adults using different methodological approaches; 2) to analyse subjects´ characteristics that could contribute to explain the differences (a) between methods used to assess central hemodynamic parameters and (b) between MBPosc and MBPc.

## Materials and methods

### Study population

In this work we considered data from a total of 1685 subjects (mean/range age: 14.4/3-35 y.o.; 854 females). The subjects´ records are part of CUiiDARTE Database, which includes data from longitudinal (cohort)and cross-sectional studies developed in Uruguay from February 2012until July 2019[10,14–16]. The subjects or their families were selected (random sampling),

mainly from their reference health institutions, educational and/or work centres, and were invited to participate through personal interviews. Included subjects were part of cohorts representative of their corresponding populations (e.g. children cohort, adolescents cohort) [16], but the whole group included in this work could not be considered (in rigorous terms) as representative of the entire Uruguayan population. Interviews, anthropometrical measurements and cardiovascular evaluations were performed in the ambulatory and/or office non-invasive vascular laboratories of CUiiDARTE. Subjects included in this work met the following criteria at the time of the evaluation: (a) all were asymptomatic and in stable clinical conditions, (b) none had congenital, chronic or infectious diseases, and (c) none was taking vasoactive drugs. Exclusion criteria included rhythm other than sinus rhythm and valvular heart disease.

All procedures agreed with the Declaration of Helsinki (1975; reviewed in 1983). The study protocol was approved by Institutional Ethic Committee (Comité de Ética en Investigación del Centro Hositalario Pereira-Rossell). Written informed consent was obtained from participants or from parents in case of subjects aged <18y.o., who gave informed assent before data collection.

## Clinical interview and anthropometric measurements

Before vascular evaluation a brief clinical interview together with the anthropometric evaluation enabled to evaluate the exposure to cardiovascular risk factors (CRFs). Subjects' body weight (BW) and height (BH) were measured and body mass index (BMI) obtained as BW-to-squared BH ratio. For subjects <18 y.o. BMI was converted into z-scores[17]. Obesity was defined as a BMI z-score $\geq 2$ for subjects aged <18 y.o. and as BMI$\geq 30$kg/m$^2$ for subjects $\geq 18$ y.o. Regular smokers (defined as usually smoking at least one cigarette per week) were identified. Sedentary life style was considered present if the subject´s physical activity (PA) was less than the recommended in terms of frequency and/or intensity[18]. To assess this, we applied questionnaires asking about characteristics of the developed PA (e.g. duration, frequency, pattern, type, intensity). Following the World Health organization (WHO) guidelines, 3 or 4 y.o. children were considered active when spending $\geq 180$ min/day in a variety of PAs that involved different intensities; of which $\geq 60$ min/day corresponded to moderate-to-vigorous PA. Subjects aged 5–17 y.o. that did not accumulate $\geq 60$ min/day of moderate-to-vigorous intensity PA were considered sedentary. The concept of accumulation refers to meeting the goal of 60 min/day by performing activities in multiple shorter bouts spread throughout the day (e.g. 2 bouts of 30 minutes). Adults ($\geq 18$ y.o.) who performed $\geq 150$ min/week of moderate-intensity aerobic PA or $\geq 75$ min/week of vigorous-intensity aerobic PA were considered active. There could be multiple ways to reach the total of 150 min/week. The concept of accumulation refers to meeting the goal of 150 min/week by performing activities in multiple shorter bouts, of $\geq 10$ minutes each, spread throughout the week. Dyslipidemia, diabetes and history of hypertension or high BP levels (HBP) was considered present if it had been previously diagnosed by referring physicians[19–21]. Subjects <16 y.o. who had pSBP and/orpDBP > 95th percentile for sex, age and BH during the study, were considered with hypertensive BP levels (disregard of previous diagnosis of hypertension). For subjects aged $\geq 16$ y.o., hypertensive BP levels were defined using cutoff values similar to those for adults (pSBP$\geq 140$ mmHg and/or pDBP$\geq 90$ mmHg)[19–21].

## Central blood pressure and wave components levels

Participants were asked to avoid exercise, tobacco, alcohol, caffeine and food-intake four hours before evaluation. All haemodynamic measurements were performed in a temperature-controlled environment (21–23˚C), with the subject in supine position and after resting for at

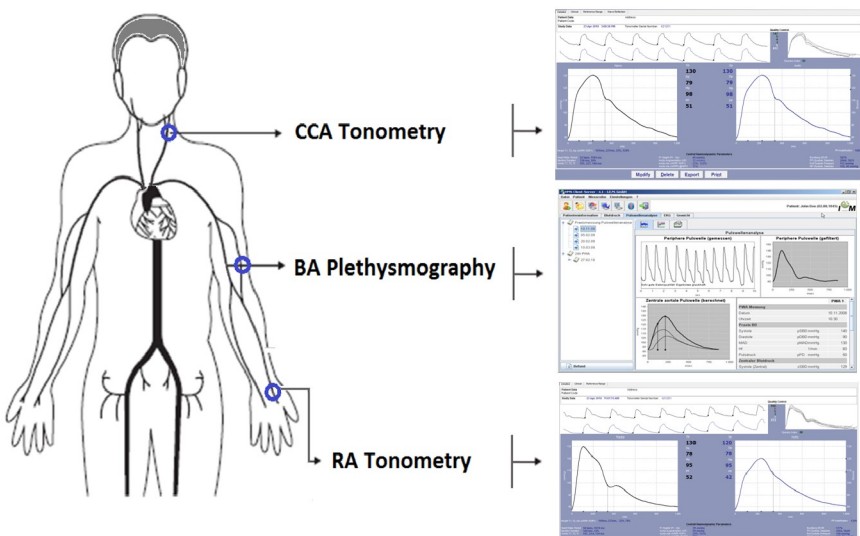

**Fig 1. Instrumental approach employed to obtain aortic blood pressure and wave components.** CCA: common carotid artery. BA: brachial artery. RA: radial artery.

least 10–15 minutes. Heart rate (HR) and brachial pSBP and pDBP were recorded in supine position using the validated oscillometric device (HEM-433INT; Omron Healthcare Inc., Illinois, USA) simultaneously and/or immediately before or after each non-invasive tonometric (radial and carotid applanation tonometry [RT and CT], respectively] and brachial oscillometry [BOSC]) recording. Peripheral pulse pressure (pPP; pPP = pSBP–pDBP) and MBPc (MBPc = pDBP+pPP/3) were obtained.

Central BP and wave components (Pf and Pb) were assessed (random order) using two commercially available devices: SphygmoCor-CvMS (SCOR; v.9, AtCor-Medical, Australia) and Mobil-O-Graph PWA-monitor system(MOG; I.E.M.-GmbH, Stolberg, Germany) [Fig 1] [9,10,11]. Both devices and systems enable doing pulse wave analysis (PWA) and wave separation analysis (WSA)[6,11,12,14].

Radial and carotid pressure waves were obtained by applanation tonometry with SCOR. The acquired waves were calibrated toMBPc and pDBP(HEM-433INT; Omron Healthcare Inc., Illinois, USA). Central BP waves were derived from radial recordings (using a GTF) and cSBP and cPP were quantified. Carotid artery pulse waves were assumed to be identical to the aortic ones (due to the proximity of the arterial sites). Thus, a GTF was not applied to obtain central waves from carotid records. Considering a triangular flow model (using WSA), Pf and Pb components of the obtained aortic waves were separated [2]. Only accurate waveforms on visual inspection and high-quality recordings (in-device quality index>75%) were considered.

Brachial BP levels and waveforms were obtained using the MOG (brachial cuff-based oscillometric device, BOSC)[21]. The device determined cBP levels and waveforms from peripheral recordings using a validated GTF. Then, by means of PWA and WSA, Pf and Pb were obtained[10,14]. Only high quality records (index equal to 1 or 2) and satisfactory waves (visual inspection) were considered. A step-by-step explanation of the method used to carry out WSA based on recorded (carotid wave, SCOR) and mathematically-derived aortic waveform (SCOR and MOG) was included as Supplementary Material (S1 Appendix). Absolute and relative intra (repeatability) and inter-observer (reproducibility) variability of cSBP, cPP, Pf and Pb was evaluated [Supplementary Material, S1 Appendix]. No significant differences were observed in cSBP, cPP, Pf or Pb absolute levels either within each visit, between two

records or between records obtained by investigators; indicating excellent repeatability, as well as reproducibility. In all cases, the relative inter- and intraobserver variability was <6%.

### Data and statistical analysis

A stepwise data analysis was done. First, ANOVA plus Bonferroni post-hoc tests were done to compare (for each age-related group) mean values obtained with the different methods (RT vs. CT vs. BOSC). Second, the relationships between cSBP, cPP, Pf and Pb data obtained with the different approaches (RT, CT and BOSC) were assessed (correlation analyses). Third, Bland-Altman analyses were performed to evaluate the agreement (equivalence) between methods. Bland-Altman plots correspond to the mean of the methods considered (x-axis; e.g. RT and CT mean) against their difference (y-axis; e.g. RT minus CT). The corresponding linear regression equations were obtained. Systematic error (bias) was considered present if mean error was significantly different from zero, whereas proportional error was considered present if the slope of the linear regression was statistically significant. Fourth, multiple linear regression models (MLR; stepwise method) were considered to analyze the association between the differences in cSBP, cPP, Pf and Pb ($\Delta$cSBP, $\Delta$cPP, $\Delta$Pf and $\Delta$Pb, respectively; absolute values) between methods [dependent variable] and age, sex, BH, BW, pDBP and HR[independent variables]. Fifth, the degree of equivalence between MBPosc and MBPc was analyzed (correlation and Bland-Altman test), and explanatory variables for the differences were identified (MBPosc minus MBPc; correlation analysis and MLR model[enter and stepwise method]). After identifying independent variables (those with p<0.01 in bivariate analysis) two different MLR models were tested. Model 1: independent variables were all significant variables in bivariate analysis (age, sex, BH, BW, BMI, HBP, pPP and pSBP; pDBP was excluded due to multicollinearity). Model 2: independent variables were pDBP and pSBP (pPP was excluded due to multicollinearity). In all MLR analyses, a variance inflation factor (VIF) <5 was selected to evaluate (discard) significant collinearity among variables.

The described analysis was done considering all the studied subjects (entire group), as well as three age-related groups: children: 3–12 y.o. (n = 728), adolescents: 12–18 y.o. (n = 361) and young adults: 18–35 y.o. (n = 596). The analysis was done calibrating peripheral signals (carotid, radial and brachial)to pDBP and MBPc data obtained at the time of signals recording ("instantaneous blood pressure"). Thereafter, the analysis was done considering carotid, radial and brachial signals calibrated to the same pressure levels, taking into account data obtained from the Mobil-O-Graph and different calibration methods: pDBP/MBPc and pDBP/MBPosc (subsample). This analysis enabled to evaluate the differences in cSBP, cPP, Pf and Pb data obtained with the different methods (RT, CT, BOSC) taking into account (i.e. with independence of)potential differences in pressure values considered for calibration and/orin MBP data used to calibrate signals (i.e. MBPc vs. MBPosc).

According to the central limit theorem, a normal distribution was considered (considering Kurtosis and Skewness coefficients distribution and number of subjects, with sample size >30) [22]. Data analyses were done using MedCalc (v.14.8.1, MedCalc Inc., Ostend, Belgium) and IBM-SPSS Statistical Software (v.20, SPSS Inc., Illinois, USA). A p value<0.05 was considered statistically significant.

## Results

### Agreement between cBP data obtained from RT, CT and BOSC

Table 1 (and S1 Table) shows characteristics of the studied subjects.

Table 2 (and S2 Table) shows peripheral and central pressure, Pf, Pb and HR levels, calibrating data to pDBP/MBPcvalues registered at the time of RT and CT assessment(device: HEM-

**Table 1. Clinical features and cardiovascular risk factors for the entire and age-related subgroups.**

| | Entire group [3–35 y.o; n = 1685] | | Children [3–12 y.o; n = 728] | | Adolescents [12–18 y.o; n = 361] | | Young adults [18–35 y.o; n = 596] | |
| --- | --- | --- | --- | --- | --- | --- | --- | --- |
| | MV | SD | MV | SD | MV | SD | MV | SD |
| Age (years) | 14.45 | 7.5 | 7.2 | 1.9 | 15.4 | 2.1 | 22.6 | 4.7 |
| Sex female, n (%) | 854 | [50.7] | 332 | [45.6] | 175 | [48.5] | 347 | [58.1] |
| Bodyheight (m) | 1.47 | 0.24 | 1.23 | 0.13 | 1.63 | 0.10 | 1.68 | 0.09 |
| Bodyweight (kg) | 49.0 | 22.6 | 29.2 | 12.9 | 61.8 | 16.5 | 66.2 | 14.1 |
| BMI (Kg./m$^2$) | 21.3 | 5.0 | 18.7 | 4.2 | 23.1 | 5.2 | 23.4 | 4.0 |
| z-BMI* (kg/m$^2$) | 1.30 | 1.88 | 1.36 | 1.88 | 1.07 | 1.88 | - | - |
| Hypertension and/or HBP, n [%] | 206 | [12.2] | 85 | [11.7] | *54* | *[15]* | 67 | [11.3] |
| Diabetes, n [%] | 7 | [0.4] | 4 | [0.6] | 3 | [0.8] | 0 | [0] |
| Dyslipidemia, n [%] | 113 | [6.7] | 35 | [4.8] | 23 | [6.4] | 55 | [9.2] |
| Obesity, n [%] | 304 | [18.7] | 204 | [28.5] | 60 | [17.6] | 40 | [7] |
| Smoking, n [%] | 145 | [8.8] | 0 | [0] | 14 | [3.9] | 131 | [23.3] |
| Family history of CV disease [%] | 0 | [0] | 0 | [0] | 0 | [0] | 0 | [0] |
| Sedentarylifestyle, n [%] | 512 | [36.7] | 159 | [22.4] | 146 | [47.9] | 207 | [54.3] |

MV: mean value. SD: standard deviation. BMI: body mass index. z-BMI:

*z-score of BMI calculated only for under 18 years old (y.o.). HBP: high blood pressure state. CV: cardiovascular.

433INT) or during cBP measurement using BOSC (device: Mobil-O-Graph, self-calibration). Within each group there was a wide-range of BP and HR values, which enabled analyzing different hemodynamic states. There was an age-related increase in pSBP, pDBP and pMBPcvalues used to calibrate RT, CT and BOSC[Table 2, Fig 2].

Tables 3 and 4 shows data from the analyses of association (correlation) and agreement (Bland-Altman) between methods used to obtain cSBP, cPP, Pfand Pb (calibration: pDBP/MBPc).

Mean error data are shown in Fig 3, ordered by means of the obtained values. Additional information about the described analyses can be found in Supplemental Material [S3–S7 Tables; S1–S4 Figs]. cSBP, cPP, Pf and Pb data from the different methodological approaches were positively associated (p<0.001) [Tables 3 and 4]. Significant mean error levels were obtained when analyzing methods´ agreement for cSBP, cPP, Pf and Pb data. The only exceptions were cSBP$_{CT-BOSC}$ in the 18–35 y.o. group (p = 0.07), Pb$_{RT-BOSC}$ in the 3–12 y.o. group (p = 0.09) and Pb$_{CT-BOSC}$ in the 12–18 y.o. group (p = 0.15) [Tables 3 and 4; Fig 3].

In turn, with few exceptions, cSBP, cPP, Pf and Pb data obtained from the different methodological approaches showed proportional errors [Tables 3 and 4; S3–S7 Tables; S1–S4 Figs]. Disregard of the age-group, for cSBP, cPP and Pf values analyzed, the greater the measurements mean, greater the differences in data from the different approaches (RT, CT, BOSC). In other words, the higher the cSBP, cPP or Pf, greater the observed differences between methods [S1 Fig].

Mean errors obtained for cSBP, cPP, Pf and Pb when comparing the methodological approaches (RT, CT and BOSC) are shown in Fig 3. Absolute values observed for mean errors ranged from1.5to9.3 mmHg for cSBP (BOSC-CT 18–35 y.o. and RT-CT 12–18 y.o.);1.6and 10.7 mmHg forcPP (RT-BOSC 18–35 y.o. and RT-CT 12–18 y.o.); 4.4and20.2 mmHg for Pf (BOSC-RT 3–12 y.o. and BOSC-CT 12–18 y.o.) and from 0.3and 2.4 mmHg for Pb (RT-BOSC 3–12 y.o. and RT-CT 18–35 y.o.) [Fig 3]. The lowest absolute values observed for mean error levels for cSBP data were obtained when analyzing CT and BOSC: 3.1 mmHg (entire group) and 1.5, 3.6 and 4.2 mmHg (young adults, adolescents and children, respectively)[Fig 3]. For

**Table 2. Haemodynamic and aortic wave-derived parameters measured with three different methods in the entire and age-related groups.**

**Entire group [3–35 y.o.; n = 1685]**

| | RT (Scor) | | CT (Scor) | | BOSC (MOG) | | P value | | |
|---|---|---|---|---|---|---|---|---|---|
| | MV | SD | MV | SD | MV | SD | RT vs CT | RT vs OSC | CT vs OSC |
| pSBP (mmHg) | 114.7 | 12.9 | 114.4 | 14.3 | 113.3 | 12.0 | 1.00 | **0.04** | 0.21 |
| pDBP (mmHg) | 64.1 | 8.7 | 62.8 | 8.1 | 63.7 | 8.0 | **0.001** | 0.88 | 0.06 |
| MBPc (mmHg) | 80.8 | 9.2 | 80.0 | 8.8 | 80.0 | 8.5 | 0.08 | 0.08 | 1.00 |
| HR (beats/minute) | 75.8 | 14.2 | 75.5 | 14.4 | 78.9 | 15.2 | 1.00 | **<0.001** | **<0.001** |
| cSBP (mmHg) | 98.6 | 12.3 | 105.9 | 15.2 | 99.7 | 14.4 | **<0.001** | 0.19 | **<0.001** |
| cPP (mmHg) | 33.3 | 9.8 | 43.0 | 13.7 | 34.6 | 11.6 | **<0.001** | **0.04** | **<0.001** |
| Pf (mmHg) | 31.1 | 10.0 | 43.2 | 14.2 | 23.5 | 7.8 | **<0.001** | **<0.001** | **<0.001** |
| Pb (mmHg) | 13.3 | 4.0 | 15.4 | 4.7 | 13.2 | 5.1 | **<0.001** | 1.00 | **<0.001** |

**Children [3–12 y.o.; n = 728]**

| | MV | SD | MV | SD | MV | SD | RT vs CT | RT vs OSC | CT vs OSC |
|---|---|---|---|---|---|---|---|---|---|
| pSBP (mmHg) | 104.8 | 10.0 | 104.7 | 11.7 | 106.7 | 1.0 | 1.00 | **0.01** | **0.01** |
| pDBP (mmHg) | 60.0 | 7.4 | 59.6 | 6.8 | 60.3 | 6.4 | 1.00 | 1.00 | 0.32 |
| MBPc (mmHg) | 74.7 | 7.0 | 74.6 | 7.1 | 75.5 | 1.0 | 1.00 | 0.11 | 0.12 |
| HR (beats/minute) | 85.0 | 14.0 | 84.6 | 0.8 | 86.9 | 14.0 | 1.00 | 0.25 | 0.11 |
| cSBP (mmHg) | 87.0 | 8.9 | 95.4 | 11.3 | 89.9 | 8.8 | **<0.001** | **<0.001** | **<0.001** |
| cPP (mmHg) | 24.4 | 7.3 | 35.8 | 10.5 | 28.3 | 7.1 | **<0.001** | **0.003** | **<0.001** |
| Pf (mmHg) | 24.5 | 7.6 | 35.8 | 10.3 | 19.5 | 4.9 | **<0.001** | **<0.001** | **<0.001** |
| Pb (mmHg) | 11.4 | 4.5 | 12.6 | 3.6 | 10.5 | 3.0 | **<0.001** | **0.004** | **<0.001** |

**Adolescents [12–18 y.o.; n = 361]**

| | MV | SD | MV | SD | MV | SD | RT vs CT | RT vs OSC | CT vs OSC |
|---|---|---|---|---|---|---|---|---|---|
| pSBP (mmHg) | 116.9 | 10.9 | 117.3 | 12.5 | 118.5 | 11.1 | 1.00 | 0.27 | 0.72 |
| pDBP (mmHg) | 63.1 | 7.6 | 62.3 | 7.9 | 65.6 | 7.4 | 0.60 | **<0.001** | **<0.001** |
| MBPc (mmHg) | 80.8 | 8.3 | 80.6 | 7.9 | 83.0 | 7.7 | 1.00 | **0.005** | **0.003** |
| HR (beats/minute) | 73.8 | 13.6 | 72.5 | 12.9 | 73.2 | 12.9 | 0.70 | 1.00 | 1.00 |
| cSBP (mmHg) | 100.5 | 9.9 | 109.6 | 13.8 | 107.2 | 12.1 | **<0.001** | **<0.001** | 0.09 |
| cPP (mmHg) | 36.1 | 9.0 | 47.3 | 13.4 | 40.1 | 11.7 | **<0.001** | **<0.001** | **<0.001** |
| Pf (mmHg) | 34.2 | 9.0 | 47.2 | 15.0 | 27.3 | 8.1 | **<0.001** | **<0.001** | **<0.001** |
| Pb (mmHg) | 13.3 | 3.3 | 15.9 | 4.7 | 15.3 | 5.4 | **<0.001** | **<0.001** | 0.50 |

**Young adults [18–35 y.o.; n = 596]**

| | MV | SD | MV | SD | MV | SD | RT vs CT | RT vs OSC | CT vs OSC |
|---|---|---|---|---|---|---|---|---|---|
| pSBP (mmHg) | 120.3 | 11.7 | 120.7 | 13.1 | 120.5 | 10.7 | 1.00 | 1.00 | 1.00 |
| pDBP (mmHg) | 67.5 | 8.7 | 66.0 | 8.2 | 68.4 | 8.3 | **0.01** | 0.62 | **0.003** |
| MBPc (mmHg) | 85.1 | 8.5 | 84.2 | 8.2 | 85.4 | 8.3 | 0.29 | 1.00 | 0.27 |
| HR (beats/minute) | 71.5 | 12.0 | 69.9 | 11.8 | 69.4 | 11.0 | 0.13 | 0.08 | 1.00 |
| cSBP (mmHg) | 104.5 | 10.2 | 112.1 | 14.3 | 110.4 | 12.7 | **<0.001** | **<0.001** | 0.3215 |
| cPP (mmHg) | 36.1 | 9.5 | 46.1 | 13.9 | 40.8 | 12.4 | **<0.001** | **<0.001** | **<0.001** |
| Pf (mmHg) | 33.5 | 10.0 | 45.6 | 14.0 | 27.1 | 8.3 | **<0.001** | **<0.001** | **<0.001** |
| Pb (mmHg) | 14.4 | 3.6 | 16.9 | 4.6 | 16.0 | 5.6 | **<0.001** | **<0.001** | 0.060 |

MV: mean value. SD: standard deviation. y.o.: years old. RT and CT: radial and carotid applanation tonometry (SphygmoCor device). BOSC: brachial oscillometry/plethysmography (Mobil-O-Graph device). pSBP, pDBP, MBPc: peripheral systolic, diastolic and mean (calculated) blood pressure. HR: heart rate. cSBP, cPP: central systolic and pulse blood pressure. Pf and Pb: forward and backward wave height. Significance: p<0.05 (ANOVA+Bonferroni post-hoc test).

all age-groups, the highest mean error values between cSBP data were obtained when comparing CT and RT [Fig 3].

When considering cPP, data from RT and BOSC were the ones with the greatest similarity (least absolute mean error values), 2.4 mmHg (entire group), 1.6, 2.6 and 2.9 mmHg (young adults, adolescents and children, respectively) [Fig 3]. Again, the greatest absolute difference

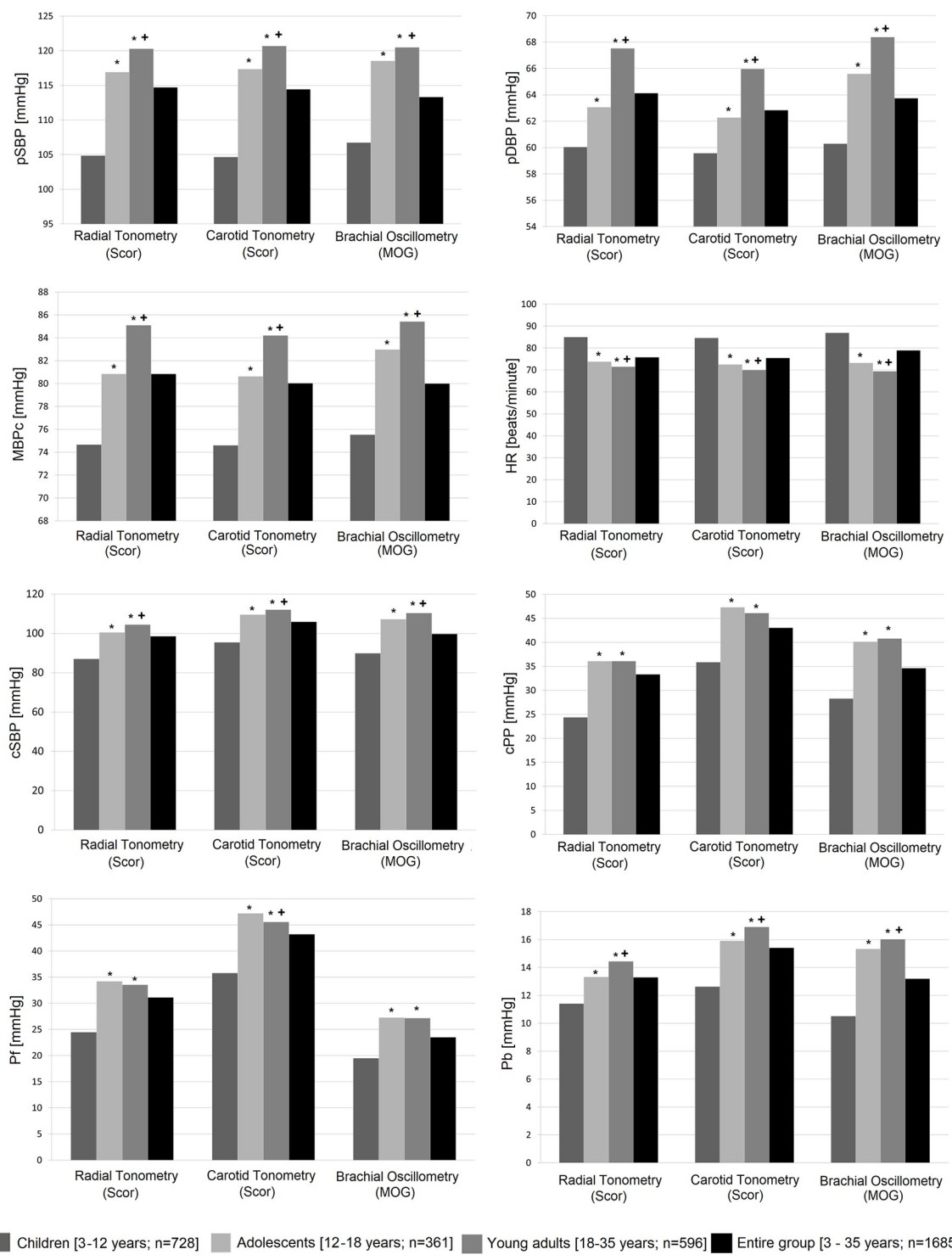

**Fig 2. Haemodynamic parameters obtained for the entire group and age-related groups.** Scor and MOG: SphygmoCor and Mobil-O-Graph. pSBP, pDBP: peripheral systolic and diastolic blood pressure. MBPc: calculated mean blood pressure. HR: heart rate. cSBP, cPP: central systolic and pulse pressure. Pf, Pb: forward and backward wave height. *p<0.05 with respect to Children; +p<0.05 with respect to Adolescents.

**Table 3. cSBP, cPP, Pf and Pb: Correlation and agreement among values obtained with three different recording methods (entire group and children).**

| | Entire group [3–35 y.o.; n = 1685] | | | Children [3–12 y.o.; n = 728] | | |
|---|---|---|---|---|---|---|
| | RT- CT | RT -BOSC | CT—BOSC | RT- CT | RT -BOSC | CT—BOSC |
| **cSBP** | | | | | | |
| R | 0.82 | 0.79 | 0.72 | 0.73 | 0.64 | 0.50 |
| P | <0.001 | <0.001 | <0.001 | <0.001 | <0.001 | <0.001 |
| ME (mmHg) | -8.0 | -5.2 | 3.1 | -7.5 | -3.9 | 4.2 |
| ME, p value | <0.001 | <0.001 | <0.001 | <0.001 | <0.001 | <0.001 |
| ME, SD (mmHg) | 8.8 | 9.2 | 11.2 | 7.1 | 7.4 | 9.8 |
| Regressionequation | y = 12.9–0.2x | y = 9.1–0.1x | y = -5.4+0.08x | y = 10.0–0.2x | y = 7.3–0.1x | y = -11.5+0.2x |
| p (Slope) | <0.001 | <0.001 | 0.03 | <0.001 | 0.038 | 0.037 |
| **cPP** | | | | | | |
| R | 0.66 | 0.64 | 0.50 | 0.57 | 0.63 | 0.46 |
| P | <0.001 | <0.001 | <0.001 | <0.001 | <0.001 | <0.001 |
| ME (mmHg) | -9.2 | -2.4 | 7.3 | -8.9 | -2.9 | 6.9 |
| ME, p value | <0.001 | <0.001 | <0.001 | <0.001 | <0.001 | <0.001 |
| ME, SD (mmHg) | 9.2 | 8.1 | 10.8 | 7.5 | 6.2 | 8.7 |
| Regressionequation | y = 2.6–0.3x | y = 0.9–0.09x | y = -0.5+0.2x | y = 0.7–0.3x | y = -2.1–0.03x | y = -2.0+0.3x |
| p (Slope) | <0.001 | 0.003 | <0.001 | <0.001 | 0.644 | 0.002 |
| **Pf** | | | | | | |
| R | 0.57 | 0.57 | 0.35 | 0.54 | 0.62 | 0.44 |
| P | <0.001 | <0.001 | <0.001 | <0.001 | <0.001 | <0.001 |
| ME (mmHg) | -10.9 | 7.1 | 18.3 | -11.0 | 4.4 | 15.6 |
| ME, p value | <0.001 | <0.001 | <0.001 | <0.001 | <0.001 | <0.001 |
| ME, SD (mmHg) | 10.2 | 7.5 | 10.9 | 8.4 | 6.0 | 8.9 |
| Regressionequation | y = 1.9–0.4x | y = -3.5+0.4x | y = -6.2+0.7x | y = -0.3–0.4x | y = -7.9+0.6x | y = -9.2+0.9x |
| p (Slope) | <0.001 | <0.001 | <0.001 | <0.001 | <0.001 | <0.001 |
| **Pb** | | | | | | |
| R | 0.69 | 0.58 | 0.44 | 0.39 | 0.57 | 0.40 |
| P | <0.001 | <0.001 | <0.001 | <0.001 | <0.001 | <0.001 |
| ME (mmHg) | -1.9 | -0.7 | 1.0 | -0.9 | 0.3 | 1.3 |
| ME, p value | <0.001 | <0.001 | <0.001 | <0.001 | 0.093 | <0.001 |
| ME, SD (mmHg) | 3.8 | 3.8 | 4.5 | 3.5 | 2.8 | 3.6 |
| Regressionequation | y = 3.8–0.4x | y = 4.9–0.4x | y = 3.2–0.1x | y = 2.8–0.3x | y = 1.7–0.1x | y = 1.5–0.02x |
| p (Slope) | <0.001 | <0.001 | 0.006 | 0.001 | 0.078 | 0.880 |

y.o.: years old. RT, CT: radial and carotid tonometry (SphygmoCor), respectively. BOSC: brachial oscillometry/plethysmography (Mobil-O-Graph). cSBP, cPP: central systolic and pulse pressure, respectively. Pf, Pb: forward and backward wave amplitude, respectively. R: correlation (Pearson) coefficient. ME: mean or systematic error. β: slope of regression equation. Significance: p<0.05. Bland-Altman: "x" was considered the mean of both methods compared (e.g. (RT+CT)/2); "y" the difference among first and second method (e.g. RT minus CT).

was observed when comparing RT and CT data (e.g. 9.2 mmHg, entire group) [Fig 3]. For all age-groups, Pf values obtained with RT and BOSC showed the most similitude (e.g. mean error 7.1 mmHg, entire group), whereas the major absolute errors were obtained when comparing BOSC and CT (e.g. 18.3 mmHg, entire group) [Fig 3]. When analyzing the entire group, the least differences in Pb were observed between BOSC and tonometry-derived data (e.g. 1.0 mmHg), whereas RT and CT data showed the greatest differences (e.g. 1.9 mmHg). When considering the different age-groups, findings were heterogeneous (e.g. in children the absolute difference in Pb between RT and CT was just 0.9 mmHg) [Fig 3].

**Table 4. cSBP, cPP, Pf and Pb: Correlation and agreement among values obtained with three different recording methods (adolescents and young adults).**

| | Adolescents [12–18 y.o.; n = 361] | | | Young adults [18–35 y.o.; n = 596] | | |
|---|---|---|---|---|---|---|
| | RT- CT | RT—BOSC | CT—BOSC | RT- CT | RT—BOSC | CT—BOSC |
| **cSBP** | | | | | | |
| R | 0.70 | 0.60 | 0.53 | 0.68 | 0.47 | 0.49 |
| P | <0.001 | <0.001 | <0.001 | <0.001 | <0.001 | <0.001 |
| ME (mmHg) | -9.3 | -6.5 | 3.6 | -7.4 | -5.5 | 1.5 |
| ME, p value | <0.001 | <0.001 | <0.001 | <0.001 | <0.001 | 0.07 |
| ME, SD (mmHg) | 9.80 | 9.89 | 12.6 | 9.9 | 10.3 | 11.0 |
| Regressionequation | y = 30.6–0.4x | y = 15.4–0.2x | y = -21.6+0.2x | y = 26.8–0.3x | y = 9.2–0.1x | y = -17.7+0.2x |
| p (Slope) | <0.001 | 0.003 | 0.004 | <0.001 | 0.10 | 0.045 |
| **cPP** | | | | | | |
| R | 0.67 | 0.47 | 0.49 | 0.65 | 0.62 | 0.52 |
| P | <0.001 | <0.001 | <0.001 | <0.001 | <0.001 | <0.001 |
| ME (mmHg) | -10.7 | -2.6 | 8.6 | -8.3 | -1.6 | 6.5 |
| ME, p value | <0.001 | <0.001 | <0.001 | <0.001 | 0.015 | <0.001 |
| ME, SD (mmHg) | 10.0 | 9.0 | 12.5 | 9.8 | 9.1 | 11.0 |
| Regressionequation | y = 9.6–0.5x | y = 6.5–0.2x | y = -7.7+0.4x | y = 6.4–0.4x | y = 5.0–0.2x | y = 5.2+0.03x |
| p (Slope) | <0.001 | 0.001 | <0.001 | <0.001 | 0.012 | 0.724 |
| **Pf** | | | | | | |
| R | 0.57 | 0.59 | 0.44 | 0.60 | 0.58 | 0.47 |
| P | <0.001 | <0.001 | <0.001 | <0.001 | <0.001 | <0.001 |
| ME (mmHg) | -11.8 | 8.3 | 20.2 | -10.3 | 9.1 | 19.0 |
| ME, p value | <0.001 | <0.001 | <0.001 | <0.001 | <0.001 | <0.001 |
| ME, SD (mmHg) | 11.0 | 7.7 | 12.2 | 10.7 | 7.9 | 10.8 |
| Regressionequation | y = 10.4–0.6x | y = 2.2+0.2x | y = -9.8+0.8x | y = 4.9–0.4x | y = 1.4+0.3x | y = -5.1+0.7x |
| p (Slope) | <0.001 | 0.007 | <0.001 | <0.001 | <0.001 | <0.001 |
| **Pb** | | | | | | |
| R | 0.56 | 0.56 | 0.40 | 0.57 | 0.54 | 0.41 |
| P | <0.001 | <0.001 | <0.001 | <0.001 | <0.001 | <0.001 |
| ME (mmHg) | -2.2 | -1.6 | 0.6 | -2.4 | -0.9 | 1.2 |
| ME, p value | <0.001 | <0.001 | 0.15 | <0.001 | 0.002 | 0.002 |
| ME, SD (mmHg) | 3.8 | 4.1 | 4.9 | 3.8 | 4.3 | 4.9 |
| Regressionequation | y = 4.3–0.4x | y = 5.1–0.5x | y = 2.4–0.1x | y = 3.5–0.4x | y = 7.3–0.6x | y = 5.5–0.3x |
| p (Slope) | <0.001 | <0.001 | 0.283 | <0.001 | <0.001 | 0.008 |

y.o.: years old. RT and CT: radial and carotid tonometry (SphygmoCor), respectively. BOSC: brachial oscillometry/plethysmography (Mobil-O-Graph). cSBP, cPP: central systolic and pulse pressure, respectively. Pf, Pb: forward and backward wave amplitude, respectively. R: correlation (Pearson) coefficient. β: slope of regression equation. ME: mean or systematic error. Significance: p<0.05. Bland-Altman: "x" was considered the mean of both methods compared (e.g. (RT+CT)/2); "y" the difference among first and second method (e.g. RT minus CT).

When data were calibrated to identical pDBP and MBP, and regardless of the calibration method[S8 and S9 Tables], Bland-Altman analyses [S10–S19 Tables; S5–S12 Figs] showed that beyond some changes in the "ranking" of the comparisons in the different age-groups, there were no substantial changes when analyzing the entire population[S13–S16 Figs]. In this regard, even when the statistical significance of the comparisons was lost, minor mean errors were observed between: (1) BOSCT-CT for cSBP; (2) RT-BOSC for cPP; (3) BOSC-RT for Pf and (4) BOSC-CT for Pb. Additionally, even when calibrating to identical values and considering calibration methods, the greatest absolute errors were still observed between: (1) RT-CT

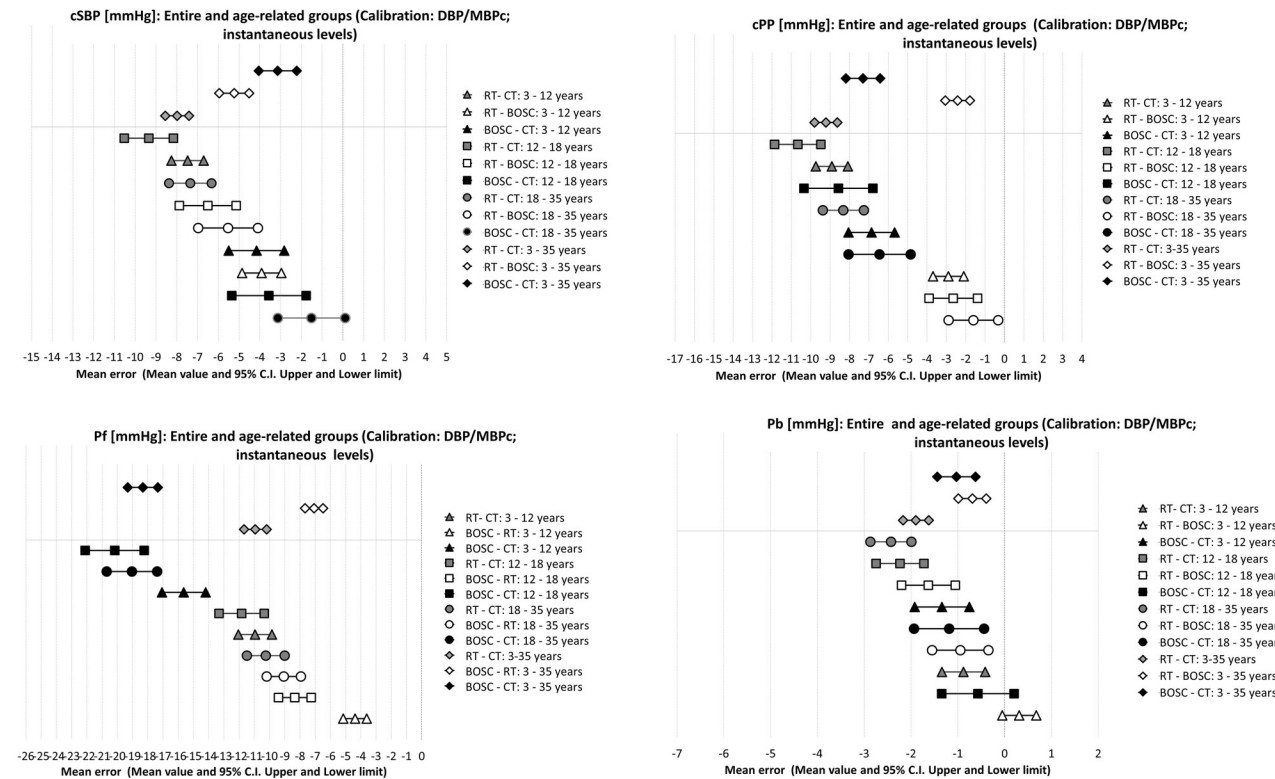

**Fig 3. Systematic (mean) error obtained in Bland-Altman test, reported as mean value and its confidence interval (95%).** RT and CT: radial and carotid applanation tonometry. BOSC: brachial oscillometry. cSBP, cPP: central (aortic) systolic and pulse pressure. Pf, Pb: forward and backward wave height.

for cSBP; (2) RT-CT for cPP; (3) BOSC-CT for Pfand (4) RT-CT for Pb (without statistical significance compared to RT-BOSC) [S13–S16 Figs].

Fig 4 (S9 Table) shows that generally when calibrating to identical pressure levels and regardless of the calibration method, the highest cSBP, cPP and Pf values were obtained from CT whilst the lowest cSBP and cPP levels were observed when using RT. Independently of the methodological approach (RT, CT or BOSC), the parameter(cSBP, cPP, Pf and Pb) and the age, higher levels were obtained when calibrating top DBP/MBPosc [Fig 4, S9 Table].

## Explanatory factors for the differences in cSBP, cPP, Pfand Pb data obtained with RT, CT and BOSC

For a given variable, the explanatory factors for the absolute differences between methods used in its measurement varied, depending on the methodological approaches that were compared (e.g. RT vs. CT and RTvs. BOSC) (Tables 5 and 6). Factors explaining the differences in cSBP and cPP data varied depending on the methods considered. In general terms, the differences in cSBP or cPP were associated with sex (major differences in males), age (negatively), pDBP (negatively), BH and/or BW (positively). Depending on the methods compared, HR was positively or negatively associated with the differences in cSBP or cPP [Tables 5 and 6]. Absolute differences in Pf were associated with age (negatively), sex (higher differences in males), BW (positively) and/or HR (positively) [Tables 5 and 6]. The differences in Pb were mainly explained by age (positively; just for RT-CT pDBP/MBPc comparison), sex (higher difference

## Entire group [3 - 35 years]

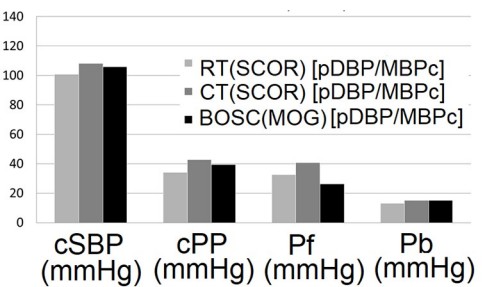 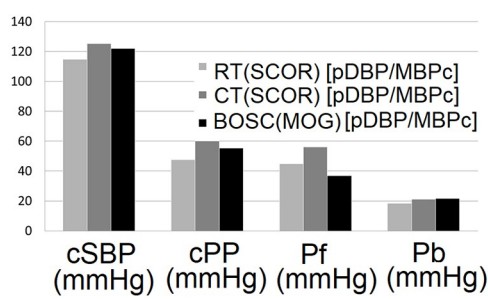

## Children [3 - 12 years]

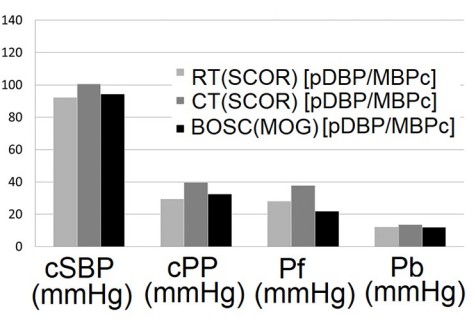 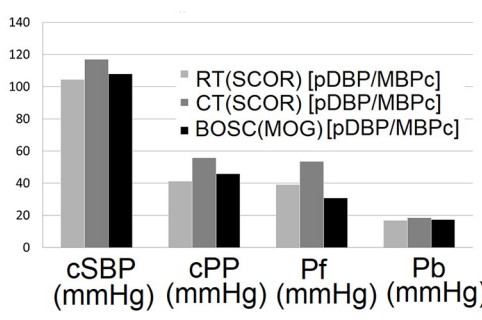

## Adolescents [12 - 18 years]

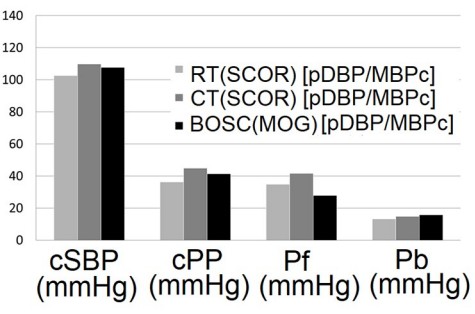 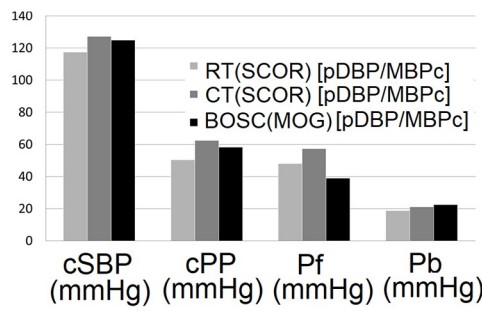

## Young adults [18 - 35 years]

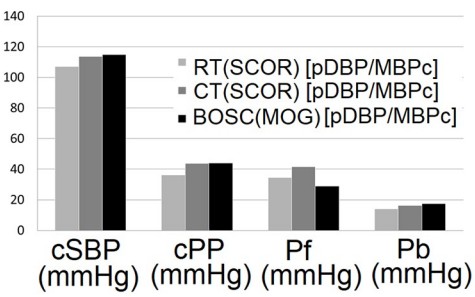 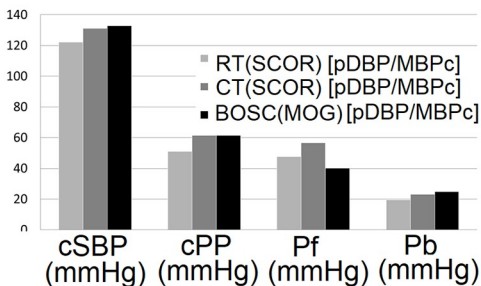

**Fig 4. Haemodynamic parameters obtained for the entire and age-related groups, when calibrating to peripheral diastolic (pDBP) and calculated mean blood pressure (MBPc) or measured mean blood pressure (MBPosc).** Scor and MOG: SphygmoCorandMobil-O-Graph device. RT and CT: radial and carotid applanation tonometry. BOSC: brachial oscillometry. cSBP, cPP: central (aortic) systolic and pulse pressure. Pf, Pb: forward and backward wave height.

**Table 5. Multiple regression models for absolute differences in cSBP and cPP levels measured with three different methods in the entire group, calibrated with identical pressure levels: pDBP/MBPc and pDBP/MBPosc.**

| | Calibration method | Variable | β | p-value (β) | VIF | R² | R²a | p-value (model) |
|---|---|---|---|---|---|---|---|---|
| **\|ΔcSBP\|** | | | | | | | | |
| RT-CT | pDBP/MBPc | Sex | -2.14 | 0.003 | 1.06 | 0.04 | 0.04 | 0.003 |
| | | HR | 0.06 | 0.016 | 1.06 | | | |
| | pDBP/MBPosc | Age | -0.45 | <0.001 | 1.47 | 0.05 | 0.04 | 0.014 |
| | | BodyWeight | 0.08 | 0.012 | 1.47 | | | |
| RT-BOSC | pDBP/MBPc | BodyHeight | 10.21 | <0.001 | 1.25 | 0.25 | 0.24 | <0.001 |
| | | pDBP | -0.18 | 0.003 | 1.13 | | | |
| | | HR | -0.25 | <0.001 | 1.14 | | | |
| | pDBP/MBPosc | BodyHeight | 12.86 | 0.001 | 1.25 | 0.27 | 0.26 | <0.001 |
| | | pDBP | -0.24 | 0.002 | 1.13 | | | |
| | | HR | -0.35 | <0.001 | 1.14 | | | |
| CT-BOSC | pDBP/MBPc | Sex | -2.64 | 0.008 | 1.00 | 0.03 | 0.02 | 0.008 |
| | pDBP/MBPosc | Sex | -3.35 | 0.014 | 1.00 | 0.02 | 0.02 | 0.014 |
| **\|ΔcPP\|** | | | | | | | | |
| RT-CT | pDBP/MBPc | Age | -7.91 | 0.012 | 2.29 | 0.13 | 0.11 | <0.001 |
| | | BodyWeight | 0.09 | <0.001 | 2.19 | | | |
| | | Sex | -2.63 | <0.001 | 1.06 | | | |
| | | HR | 0.12 | <0.001 | 1.16 | | | |
| | pDBP/MBPosc | Sex | -3.76 | <0.001 | 1.06 | 0.09 | 0.08 | <0.001 |
| | | HR | 0.18 | <0.001 | 1.06 | | | |
| RT-BOSC | pDBP/MBPc | BodyHeight | 10.26 | <0.001 | 1.25 | 0.25 | 0.24 | <0.001 |
| | | pDBP | -0.20 | <0.001 | 1.13 | | | |
| | | HR | -0.21 | <0.001 | 1.14 | | | |
| | pDBP/MBPosc | BodyHeight | 14.43 | <0.001 | 1.01 | 0.11 | 0.10 | <0.001 |
| | | Sex | -3.94 | 0.001 | 1.01 | | | |
| CT-BOSC | pDBP/MBPc | Sex | -2.44 | 0.009 | 1.00 | 0.03 | 0.02 | 0.009 |
| | pDBP/MBPosc | Sex | -3.58 | 0.005 | 1.06 | 0.04 | 0.03 | 0.008 |
| | | HR | 0.10 | 0.042 | 1.06 | | | |

cSBP, cPP: central systolic and pulse blood pressure, respectively. HR: heart rate. RT and CT: radial and carotid, tonometry, respectively, obtained with SphygmoCor device (SCOR). BOSC: brachial oscillometry obtained with Mobil-O-Graph device (MOG). |ΔcSBP|, |ΔcPP|: absolute values of difference of cSBP or cPP obtained by resting the two methods of measurement as are shown in columns. For linear regression models the dependent variable were ΔcSBP and ΔcPP, while independent variables were age (y.o.), body height (m), body weight (Kg.), sex (female: 1, male:0), peripheral (brachial) diastolic blood pressure (pDBP, mmHg) and heart rate (HR, beats/minute) entered with stepwise method. MBPc: mean blood pressure calculated as pDBP+(pSBP-pDBP*1/3). MBPosc: mean blood pressure measured by oscillometry. β: slope of regression equation. VIF: variance inflation factor. R²a: adjusted R². Significance level: p value <0.05.

for males), HR and/or pDBP (negatively) and/or BH (positively). Differences in Pf were explained by BW, rather than by BH; the opposite was observed when analyzing Pb.

## Explanatory factors for the differences between MBPosc and MBPc

MBPosc and MBPc showed a strong positive association (R = 0.9887; p<0.0001). MBPosc values were higher than MBPc levels. Systematic (6.9±1.3 mmHg, p = 0.0003) and proportional errors were observed. The differences increased in association with higher MBP levels (p = 0.0003) [S20 Table, S17 Fig]. Table 7 (bivariate models) and Table 8 (MLR) show explanatory variables for the differences between MBPosc and MBPc.

**Table 6. Multiple regression models for absolute differences in Pf or Pb levels measured with three different methods in the entire group, calibrated with identical pressure levels: pDBP/MBPc and pDBP/MBPosc.**

| | Calibration method | Variable | B | p-value (β) | VIF | $R^2$ | $R^2a$ | p-value (model) |
|---|---|---|---|---|---|---|---|---|
| **\|ΔPf\|** | | | | | | | | |
| RT-BOSC | pDBP/MBPc | Age | -0.20 | 0.011 | 1.43 | 0.11 | 0.10 | <0.001 |
| | | BodyWeight | 0.11 | <0.001 | 1.43 | | | |
| | pDBP/MBPosc | Age | -0.29 | 0.007 | 1.43 | 0.11 | 0.10 | <0.001 |
| | | BodyWeight | 0.14 | <0.001 | 1.43 | | | |
| CT-BOSC | pDBP/MBPc | BodyWeight | 0.09 | 0.001 | 1.03 | 0.14 | 0.13 | <0.001 |
| | | Sex | -2.87 | 0.010 | 1.08 | | | |
| | | HR | 0.21 | <0.001 | 1.11 | | | |
| | pDBP/MBPosc | Age | -0.49 | 0.006 | 1.45 | 0.10 | 0.06 | <0.001 |
| | | BodyWeight | 0.15 | 0.002 | 1.37 | | | |
| | | HR | 0.15 | 0.022 | 1.09 | | | |
| **\|ΔPb\|** | | | | | | | | |
| RT-CT | pDBP/MBPc | Age | 0.05 | 0.025 | 1.10 | 0.09 | 0.08 | <0.001 |
| | | HR | -0.03 | 0.007 | 1.10 | | | |
| | pDBP/MBPosc | BodyHeight | 2.09 | 0.047 | 1.10 | 0.08 | 0.07 | 0.001 |
| | | HR | -0.04 | 0.016 | 1.10 | | | |
| RT-BOSC | pDBP/MBPc | BodyHeight | 6.72 | <0.001 | 1.24 | 0.26 | 0.25 | <0.001 |
| | | pDBP | -0.10 | <0.001 | 1.13 | | | |
| | | HR | -0.09 | <0.001 | 1.13 | | | |
| | pDBP/MBPosc | BodyHeight | 10.57 | <0.001 | 1.25 | 0.31 | 0.30 | <0.001 |
| | | Sex | -1.19 | 0.049 | 1.10 | | | |
| | | pDBP | -0.12 | <0.001 | 1.15 | | | |
| | | HR | -0.11 | <0.001 | 1.19 | | | |
| CT-BOSC | pDBP/MBPc | BodyHeight | 3.67 | 0.007 | 1.13 | 0.14 | 0.13 | <0.001 |
| | | HR | -0.07 | <0.001 | 1.13 | | | |
| | pDBP/MBPosc | BodyHeight | 5.45 | <0.001 | 1.12 | 0.13 | 0.12 | <0.001 |
| | | HR | -0.06 | 0.013 | 1.12 | | | |

Pf, Pb: forward and backward wave height (amplitude) at aortic level, respectively. HR: heart rate. RT and CT: radial and carotid, tonometry, respectively, obtained with SphygmoCor device (SCOR). BOSC: brachial oscillometry obtained with Mobil-O-Graph device (MOG). |ΔPf|, |ΔPb|: absolute values of difference of Pf and Pb obtained by resting the two methods of measurement as are shown in columns. For linear regression models the dependent variable were ΔPf or ΔPb, while independent variables were age (y.o.), body height (m), body weight (Kg.), sex (female: 1, male:0), peripheral (brachial) diastolic blood pressure (pDBP, mmHg) and heart rate (HR, beats/minute) entered with stepwise method. MBPc: mean blood pressure calculated as pDBP+(pSBP-pDBP*1/3). MBPosc: mean blood pressure measured by oscillometry. β: slope of regression equation. VIF: variance inflation factor. $R^2a$: adjusted $R^2$. Significance level: p value <0.05.

The differences between MBPosc and MBPc were associated with sex (major differences in males), pDBP (negatively) and age, BH, BW, BMI, z-BMI, pSBP, pPP, hypertension and/or HBP (positively) [Table 7]. However, the multivariate analysis showed that when demographic, anthropometric, haemodynamic and CRFs variables were jointly considered (Model 1; stepwise and enter), the variable with major explanatory capacity was pPP (positive association), but when considering the enter method, the associations for age (positive, p = 0.049) and BH (negative, p = 0.030) were statistically significant (p<0.05). The $R^2$ showed little variation when considering the three variables (pPP, age, BH) in the equation(0.880 for pPP vs. 0.883 for pPP, age and BH). When analyzing pDBP, pSBP and pPP (Model 2), it was observed that the differences between MBPosc and MBPc were associated with pSBP (positive association)and with pDBP (negative association) [Table 8].

**Table 7. Differences between MBPosc and MBPc: association with demographic data, anthropometric data, risk factors and haemodynamic properties.**

| | MBPosc—MBPc (mmHg) | |
| --- | --- | --- |
| **Demographic and anthropometric variables** | **R** | **P** |
| Age (years) | 0.165 | 0.007 |
| Sex (1: female, 0: male) | -0.247 | <0.0001 |
| Bodyheight (m) | 0.283 | <0.0001 |
| Bodyweight (kg) | 0.365 | <0.0001 |
| BMI (m/kg$^2$) | 0.297 | <0.0001 |
| z-BMI* (standard deviation) | 0.276 | 0.001 |
| **Cardiovascular RiskFactors** | **R** | **P** |
| Hypertension and/or HBP [1: yes, 0: no] | 0.179 | 0.005 |
| Diabetes [1: yes, 0: no] | -0.045 | 0.48 |
| Dyslipidemia[1: yes, 0: no] | -0.008 | 0.90 |
| Smoking[1: yes, 0: no] | 0.073 | 0.25 |
| Sedentarylifestyle[1: yes, 0: no] | 0.046 | 0.48 |
| **Haemodynamic parameters** | **R** | **P** |
| HR (beats/minute) | -0.099 | 0.11 |
| pDBP (mmHg) | -0.205 | <0.001 |
| pSBP (mmHg) | 0.653 | <0.0001 |
| pPP (mmHg) | 0.934 | <0.0001 |

MBP: mean blood pressure. BMI: body mass index. HBP: high blood pressure. HR: heart rate. pDBP, pSBP, pPP: peripheral diastolic, systolic and pulse pressure. R: pearson coefficient. Significance level: p-value <0.05.

*z-BMI: z score of BMI calculated only for under 18 y.o.

**Table 8. Multiple regression models for differences between MBPosc and MBPc: Association with respect to demographic, anthropometric, risk factors and haemo-dynamic properties.**

| Dependent variable: MBPosc-MBPc | Variable | β | p-value (β) | VIF | R$^2$ | R$^2$a | p-value (model) |
| --- | --- | --- | --- | --- | --- | --- | --- |
| Model 1 (Enter method) | Age | 0.019 | 0.049 | 2.69 | 0.883 | 0.880 | <0.001 |
| | Sex | -0.078 | 0.223 | 1.16 | | | |
| | BodyHeight | -0.610 | 0.030 | 2.82 | | | |
| | BMI | 0.008 | 0.220 | 1.27 | | | |
| | pSBP | 0.003 | 0.444 | 2.41 | | | |
| | pPP | 0.124 | <0.001 | 2.03 | | | |
| Model 1 (Stepwise method) | pPP | 0.127 | <0.001 | 1.00 | 0.880 | 0.879 | <0.001 |
| Model 2 (Enter method) | pSBP | 0.127 | <0.001 | 1.42 | 0.874 | 0.873 | <0.001 |
| | pDBP | -0.121 | <0.001 | 1.42 | | | |
| Model 2 (Stepwise method) | pSBP | 0.127 | <0.001 | 1.42 | 0.874 | 0.873 | <0.001 |
| | pDBP | -0.121 | <0.001 | 1.42 | | | |

BMI: body mass index. pDBP, pSBP, pPP: peripheral (brachial) diastolic, systolic and pulse pressure. HTA and HBP: Hypertension and/or high blood pressure. MBPc: mean blood pressure calculated as pDBP+(pSBP-pDBP*1/3). MBPosc: mean blood pressure measured by oscillometry [mmHg]. For all multiple linear regression models the difference between MBPosc and MBPc was the dependent variable. Model 1: independent variables were age [y.o], sex(female: 1, male: 0), body height(m), body weight(Kg.), body mass index (BMI; Kg./m$^2$), HTA/HBP(yes: 1, no: 0), pDBP, pPP and pSBP(mmHg). Body weight, HTA/HBPand pDBP; pDBPwas excluded due to multicollinearity defined as variance inflation factor (VIF)>5. Model 2: independent variables were pDBP, pSBP, and pPP;pPP was excluded due to multicollinearity. β: slope of regression equation. Significance level: p value <0.05.

## Discussion

Our main results were: *First*, systematic and proportional errors were observed when analyzing methods agreement. Statistical significance and errors values varied according to the parameter analyzed, the age group considered and the methods compared. When analysing cSBP and cPP data the methods with the greatest similarity varied, but for both variables the greatest differences were obtained when comparing RT and CT data [Tables 3 and 4, Fig 3].

*Second*, with few exceptions, for cSBP, cPP, Pf or Pb data the methodological approaches with major and least similarities did not vary in association with variations in the calibration scheme considered (pDBP/MBPc vs. pDBP/MBPosc) [Tables 3 and 4, Fig 3]. Regardless of the calibration scheme when data were calibrated to similar pBP, the highest cSBP, cPP and Pf levels were obtained from CT, whereas the lowest cSBP and cPP values were obtained from RT [Fig 4, S9 Table]. Disregard of the technique (RT, CT or BOSC), parameter (cSBP, cPP, Pf or Pb) or age group (children, adolescents or young adults) considered, higher absolute values were obtained when data were calibrated to pDBP/MBPosc [Fig 4, S9 Table].

*Third*, age, sex, HR, pDBP, BW and BH were explanatory factors for the differences in cSBP, cPP, Pf or Pb [Tables 5 and 6]. Considering a given central parameter (e.g. cPP), explanatory factors for the differences between data obtained from different approaches would vary depending on the methods compared (e.g. RT vs. CT and RT vs. BOSC) [Tables 5 and 6]. Regardless of demographic (age, sex) and anthropometric (BW, BH and BMI) variables and CRFs exposure (e.g. HBP), pPP, pSBP and pDBP, were explanatory factors for the difference between MBPosc and MBPc [Table 8].

Our results showed that the greatest differences were observed between data obtained using a similar technique (i.e. applanation tonometry) and calibrating signals to similar BP levels. In this regard, as described, the greatest differences for cSBP and cPP data were obtained when comparing RT and CT. On the other hand, the agreement between methods varied according to the parameter studied (cSBP, cPP, Pf or Pb). Then, an adequate analysis of the equivalence between data obtained with different techniques and devices requires considering individually the different parameters, being aware that results obtained for a given variable cannot be extrapolated to others. Although there were variations in the absolute differences between data from different methods, in general terms, the "ordering" of approaches defined by the degree of agreement between data did not vary depending on the calibration method considered.

Higher cSBP, cPP and Pf levels were obtained with CT, whereas minor cSBP and cPP values were obtained with RT. Then, for the methodological approaches analyzed it could be said that, the closer to the aortic root the register is achieved, the higher cSBP and cPP levels obtained. This issue could be explained by the differences in pBP measured invasive and non-invasively[23]. About this, regardless of the technique used (e.g. oscillometric or auscultatory) cuff BP under-estimated intra-arterial pSBP (and pPP), at the same time it over-estimated intra-arterial pDBP[24]. Consequently, when using pDBP and MBP (calculated or measured) to calibrate arterial signals (e.g. carotid, brachial or radial), if pSBP is obtained first (and thereafter cSBP), greater errors or differences (underestimation) could be expected when considering peripheral arteries records[8,23].

For the different methodological approaches, parameters and age-groups, the highest absolute levels were obtained when calibrating to pDBP/MBPosc [Fig 4, S9 Table]. This finding is in agreement with the expected. Compared to other calibration methods (i.e. pSBP/pDBP, pDBP/MBPc), calibrating to MBPosc (pDBP/MBPosc) enables obtaining higher cSBP levels which in turn would be more alike those measured invasively[3]. On the other hand, in works that mostly used BOSC (i.e. Mobil-O-Graph) in adults, it was shown that MBPosc would be closer to the true invasive MBP and therefore compared to MBPc, the MBPosc would be more

accurate to calibrate signals[25]. cSBP obtained calibrating to DBP/MBPosc has shown: (1) better correlation with cardiac hypertrophy when using 24-hour ambulatory cSBP data[26], (2) superior discriminatory capability, associated with significant improvement in reclassification to identify cardiac structural abnormalities in community-based patients with stage A heart failure[27], (3) association with clinical outcomes in patients with chronic kidney disease[28], (4) enhanced association with cardiac structural features in children, adolescents and young adults [29]. Additionally, comparatively, cSBP obtained calibrating to DBP/MBPosc showed a weaker association with pSBP, which would increase the independent predictive capacity [29,30,31]. The value of considering cSBP apart from pSBP has been discussed and considered over the last two decades. It has been stated that their association a-priori limits the incremental clinical value of cSBP[32]. However, it has been demonstrated that the association could be modified by the measurement procedure and particularly by the calibration method [8,27,28,33]. By using pSBP (i.e. within the MBPc equation) as a direct input variable for cSBP estimation, an intrinsic mathematical connection is established systematically and an association between variables is predetermined. Therefore, cSBP obtained using oscillometric data for calibration, avoids the use of pSBP and considers measured MBP (MPBosc) instead. Then, the impact of pSBP is attenuated[8,27,28,33,34]. In this work, we showed for the first time in children and adolescents, that cSBP levels obtained calibrating to MBPosc would be higher than those obtained when calibrating to MBPc (Form factor: 0.33), regardless of the technique considered: RT, CT, BOSC.

For a given hemodynamic variable (e.g. cPP) the explanatory factors for the differences between data obtained with the different approaches varied depending on the methods compared and/or calibration schemes. Looking for explanation to these findings it could be proposed that the age and/or HR-association described for pulse amplification and arterial stiffness (mainly in central arteries)[13], contributes to understand the age and/or HR dependence observed for the differences in cSBP, cPP and Pf data obtained from central and peripheral tonometric records (RT-CT). Additionally, sex and BH explained the differences between some methods, which could be related: (a) with the algorithms (not disclosed) used by the devices (e.g. that could incorporate BH for the assessment of cSBP) and/or (b) to the fact that an average GTF is used for all subjects. Regarding the latter, some devices (e.g. SphygmoCor) frequently use an average GTF (a frequency-dependent transformation) to correlate measured radial BP waves to measured cBP waves, obtained in a group of subjects. Then, the GTF is applied to radial records from new subjects to estimate cBP waves and parameters. Despite it has been demonstrated that GTF could yield good agreement with invasive cBP measurements, since the GTF is a population average, it could assume that PP amplification is just a fixed value. Hence, the GTF may not adapt to the aforesaid inter-subject variability in PP amplification and therefore it could yield non-trivial cBP errors when PP amplification is non-uniform [35]. In this sense, since in general males show greater center-periphery amplification of the pulse and/or present greater BH than females, the use of a single GTF could explain the differences found between methods. Consequently, in addition to the "direct" effects that biological factors (e.g. subjects´ characteristics) may have on the methods and records (e.g. CT limitations in obese subjects, children or women) that would determine or contribute to the differences between data from different methods, they could also be integrated into the equations (usually not given to the users) used by the devices to obtain hemodynamic parameters. Therefore, biological factors could also contribute "indirectly" (mathematically) to the differences between data from different methodological approaches.

Our data suggest that the differences between MBPosc and MBPc [S20 Table, S17 Fig], that contribute to explain differences in central parameters obtained calibrating to pDBP/MBPosc or to pDBP/MBPc were not associated with demographic or anthropometric variables, nor

with HR. On the other hand, they were associated with pSBP, pPP or pDBP levels [Table 8]. Higher pSBP (or pPP) and lower pDBP levels were associated with higher differences between MBPosc and MBPc. This is in agreement with Kiers et al. [36], who after bivariate and multivariate analysis (data from subjects without cardiovascular disease) reported that the differences between MBPosc and MBPc were not associated with sex, age, BMI or HR, but with pSBP and pDBP. In turn, Bos et al. showed that MBPc (MBPc = pDBP+1/3pPP) underestimates "real" MBP (invasively measured), with larger underestimations at higher pressure levels [37]. That indirectly suggests that if MBPosc really approximates more to "real" MBP than MBPc, then the greater the pressure levels, the greater the MBPosc-MBPc difference. The result of the univariate analysis is in agreement with that reported by Smulyan et al. who evaluated the difference between MBPc and MBPosc (patients who underwent a coronary angiography), and observed an association ("a weak correlation") between MBP differences and age (r = 0.32; p<0.001); unfortunately, the authors did not perform multivariate analysis including blood pressure, which precludes a full comparison [25].

## Strengths and limitations

Due to the characteristics of the studied population invasive data were not obtained. However, characteristics of the studied sample comply with what was stated by Sharman et al. (2017): participants should have a sex distribution of at least 30% male and female, in sinus rhythm; devices should be tested over a range of BP and across a range of HRs (i.e. 60–100 b.p.m.)[9]. In this study, females were within 46–58% of the entire population and age-related groups, all the subjects were in sinus rhythm, pSBP and pDBP range: 70–217 mmHg and 42–106 mmHg, respectively, and HR range was 42–151 beats/minute. We did not measure MBP invasively, so we cannot conclude on the best way to quantify non-invasively MBP to be used for calibration. On the other hand, because the measurement algorithm used by the oscillometric device is unknown, we cannot explain the differences between oscillometric and calculated MBP levels as derived from the algorithm. In spite of this, the comparative analysis of MBP has important practical implications. Researchers using an oscillometric device for obtaining MBP should be aware of the differences between calculated and measured MBP. Additionally, researchers should describe the method used to obtain MBP with an oscillometric device precisely and whether measured or calculated MBP was used should be specially indicated. We do want to underline the importance of describing the method used to determine MBP when using an oscillometric device. A problem arises when oscillometric devices (e.g. the Mobil-O-Graph) do not report (in the display) MBPosc. Consequently, users usually calculate MBP from pSBP and pDBP values given by the device. It is clear that despite they are calculated using the same device and record, MBPosc and MBPc (form factor equal to 0.33) obtained are not similar. Furthermore, it is to note that there are other methods to quantify MBPc that we did not analyze since they are not used by methodological approaches considered in this work (Mobil-O-Graph).

## Conclusions

Systematic and proportional errors were observed. When analysing cSBP and cPP, there were differences in the techniques with the greatest similarity, but for both variables the greatest differences were obtained when comparing RT and CT data. In general terms, for cSBP, cPP, Pf or Pb data the methods (RT, CT and BOSC) with major and least similarities did not vary in association with variations in the calibration scheme considered. Regardless of the calibration scheme, when data were calibrated to similar pBP, the highest cSBP, cPP and Pf levels were obtained from CT, whereas the lowest cSBP and cPP values were obtained using RT. Higher

cSBP, cPP, Pf or Pb absolute values were obtained when data were calibrated to pDBP/ MBPosc. Age, sex, HR, pDBP, BW and/or BH were explanatory factors for the differences in cSBP, cPP, Pf or Pb. For a given central parameter (cSBP, cPP, Pf, Pb) the explanatory factors for the differences between data obtained from different approaches would vary depending on the methods compared (e.g. RT vs. CT and RT vs. BOSC). pSBP and pPP (positively) and pDBP (negatively) were explanatory factors for the differences between MBPosc and MBPc.

## Supporting information

**S1 Appendix.** A. Wave separation analysis (WSA) [SphygmoCor and Mobil-O-Graph]. B. cSBP, cPP, Pf and Pb absolute and relative intra (repeatability) and inter-observer (reproducibility) variability.
(DOCX)

**S1 Fig. Bland-Altman graphs for cSBP (Calibration: pDBP/MBPc; instantaneous levels): Entire and age-related groups.**
(DOCX)

**S2 Fig. Bland-Altman graphs for cPP(Calibration: pDBP/MBPc; instantaneous levels): Entire and age-related groups.**
(DOCX)

**S3 Fig. Bland-Altman graphs for Pf(Calibration: pDBP/MBPc; instantaneous levels): Entire and age-related groups.**
(DOCX)

**S4 Fig. Bland-Altman graphs for Pb (Calibration: pDBP/MBPc; instantaneous levels): Entire and age-related groups.**
(DOCX)

**S5 Fig. Bland-Altman graphs for cSBP(Calibration: pDBP/MBPc; equal levels): Entire and age-related groups.**
(DOCX)

**S6 Fig. Bland-Altman graphs for cSBP(Calibration: pDBP/MBPosc; equal levels): Entire and age-related groups.**
(DOCX)

**S7 Fig. Bland-Altman graphs for cPP(Calibration: pDBP/MBPc; equal levels): Entire and age-related groups.**
(DOCX)

**S8 Fig. Bland-Altman graphs for cPP(Calibration: pDBP/MBPosc; equal levels): Entire and age-related groups.**
(DOCX)

**S9 Fig. Bland-Altman graphs for Pf(Calibration: pDBP/MBPc; equal levels): Entire and age-related groups.**
(DOCX)

**S10 Fig. Bland-Altman graphs for Pf(Calibration: pDBP/MBPosc; equal levels): Entire and age-related groups.**
(DOCX)

**S11 Fig. Bland-Altman graphs for Pb (Calibration: pDBP/MBPc; equal levels): Entire and age-related groups.**
(DOCX)

**S12 Fig. Bland-Altman graphs for Pb (Calibration: pDBP/MBPosc; equal levels): Entire and age-related groups.**
(DOCX)

**S13 Fig. Bland-Altman-derived cSBP Mean Error levels obtained with three different recording methods and calibration schemes: pDBP/MBPc (instantaneous levels), pDBP/ MBPc (equal levels) and pDBP/MBPosc (equal levels).**
(DOCX)

**S14 Fig. Bland-Altman-derived cPP Mean Error levels obtained with three different recording methods and calibration schemes: pDBP/MBPc (instantaneous levels), pDBP/ MBPc (equal levels) and pDBP/MBPosc (equal levels).**
(DOCX)

**S15 Fig. Bland-Altman-derived Pf Mean Error levels obtained with three different recording methods and calibration schemes: pDBP/MBPc (instantaneous levels), pDBP/ MBPc (equal levels) and pDBP/MBPosc (equal levels).**
(DOCX)

**S16 Fig. Bland-Altman-derived Pb Mean Error levels obtained with three different recording methods and calibration schemes: pDBP/MBPc (instantaneous levels), pDBP/ MBPc (equal levels) and pDBP/MBPosc (equal levels).**
(DOCX)

**S17 Fig. Bland-Altman-derived mean and systematic error levels for MBPosc and MBPc differences.**
(DOCX)

**S1 Table. Clinical features and cardiovascular risk factors for the entire and age-related subgroups (Extended table).**
(DOCX)

**S2 Table. Haemodynamic and aortic wave-derived parameters measured with three different methods in the entire and age-related subgroups.**
(DOCX)

**S3 Table. cSBP, cPP, Pf and Pb: Correlation and agreement among values obtained with three different recording methods (Summary Table).**
(DOCX)

**S4 Table. cSBP: Correlation and agreement among values obtained with three different recording methods.**
(DOCX)

**S5 Table. cPP: Correlation and agreement among values obtained with three different recording methods.**
(DOCX)

**S6 Table. Pf: Correlation and agreement among values obtained with three different recording methods.**
(DOCX)

**S7 Table. Pb: Correlation and agreement among values obtained with three different recording methods.**
(DOCX)

**S8 Table. Clinical features and cardiovascular risk factors for the entire and age-related subgroups: Subsample.**
(DOCX)

**S9 Table. Haemodynamic and aortic wave-derived parameters measured with three different methods in the entire and age-related subgroups, calibrated with identical peripheral blood pressure levels obtained by oscillometry, using two different calibration schemes: pDBP/MBPc and pDBP/MBPosc.**
(DOCX)

**S10 Table. cSBP and cPP: Agreement among parameters measured with three different methods in the entire and age-related subgroups, calibrated with identical peripheral blood pressure levels obtained by oscillometry, using two different calibration schemes: pDBP/MBPc and pDBP/MBPosc [Summary table].**
(DOCX)

**S11 Table. Pf and Pb: Agreement among parameters measured with three different methods in the entire and age-related subgroups, calibrated with identical peripheral blood pressure levels obtained by oscillometry, using two different calibration schemes: pDBP/MBPc and pDBP/MBPosc [Summary table].**
(DOCX)

**S12 Table. cSBP: Agreement among parameters measured with three different methods in the entire and age-related subgroups, calibrated with identical peripheral blood pressure levels obtained by oscillometry (Calibration scheme: pDBP/MBPc) [Extended table].**
(DOCX)

**S13 Table. cPP: Agreement among parameters measured with three different methods in the entire and age-related subgroups, calibrated with identical peripheral blood pressure levels obtained by oscillometry (Calibration scheme: pDBP/MBPc) [Extended table].**
(DOCX)

**S14 Table. Pf: Agreement among parameters measured with three different methods in the entire and age-related subgroups, calibrated with identical peripheral blood pressure levels obtained by oscillometry (Calibration scheme: pDBP/MBPc) [Extended table].**
(DOCX)

**S15 Table. Pb: Agreement among parameters measured with three different methods in the entire and age-related subgroups, calibrated with identical peripheral blood pressure levels obtained by oscillometry (Calibration scheme: pDBP/MBPc) [Extended table].**
(DOCX)

**S16 Table. cSBP: Agreement among parameters measured with three different methods in the entire and age-related subgroups, calibrated with identical peripheral blood pressure levels obtained by oscillometry (Calibration scheme: pDBP/MBPosc) [Extended table].**
(DOCX)

**S17 Table. cSBP: Agreement among parameters measured with three different methods in the entire and age-related subgroups, calibrated with identical peripheral blood pressure**

levels obtained by oscillometry (Calibration scheme: pDBP/MBPosc) [Extended table]. (DOCX)

**S18 Table. Pf: Agreement among parameters measured with three different methods in the entire and age-related subgroups, calibrated with identical peripheral blood pressure levels obtained by oscillometry (Calibration scheme: pDBP/MBPosc) [Extended table].** (DOCX)

**S19 Table. Pb: Agreement among parameters measured with three different methods in the entire and age-related subgroups, calibrated with identical peripheral blood pressure levels obtained by oscillometry (Calibration scheme: pDBP/MBPosc) [Extended table].** (DOCX)

**S20 Table. Agreement among oscillometry-derived and calculated mean blood pressure (MBPosc and MBPc, respectively) obtained with Mobil-O-Graph device (MOG).** (DOCX)

## Acknowledgments

We thank the children, adolescents, and adults, and their families, for their participation in the study.

## Author Contributions

**Conceptualization:** Pedro Chiesa, Daniel Bia, Yanina Zócalo.

**Data curation:** Daniel Bia, Yanina Zócalo.

**Formal analysis:** Agustina Zinoveev, Daniel Bia, Yanina Zócalo.

**Funding acquisition:** Daniel Bia, Yanina Zócalo.

**Investigation:** Agustina Zinoveev, Juan M. Castro, Victoria García-Espinosa, Mariana Marin, Pedro Chiesa, Daniel Bia, Yanina Zócalo.

**Methodology:** Agustina Zinoveev, Juan M. Castro, Victoria García-Espinosa, Mariana Marin, Daniel Bia, Yanina Zócalo.

**Project administration:** Daniel Bia, Yanina Zócalo.

**Resources:** Daniel Bia, Yanina Zócalo.

**Supervision:** Daniel Bia, Yanina Zócalo.

**Validation:** Daniel Bia, Yanina Zócalo.

**Visualization:** Daniel Bia, Yanina Zócalo.

**Writing – original draft:** Agustina Zinoveev, Pedro Chiesa, Daniel Bia, Yanina Zócalo.

**Writing – review & editing:** Agustina Zinoveev, Juan M. Castro, Victoria García-Espinosa, Mariana Marin, Pedro Chiesa, Daniel Bia, Yanina Zócalo.

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
