## [Decision Letter · Decision Letter 0]

18 Sep 2019

PONE-D-19-22199

Aortic pressure and forward and backward wave components in children, adolescents and young-adults: agreement among data from brachial oscillometry, and radial and carotid tonometry, and factors associated with their differences

PLOS ONE

Dear Dr. Bia,

Thank you for submitting your manuscript to PLOS ONE. After careful consideration, we feel that it has merit but does not fully meet PLOS ONE’s publication criteria as it currently stands. Therefore, we invite you to submit a revised version of the manuscript that addresses the points raised during the review process.

We would appreciate receiving your revised manuscript by Nov 02 2019 11:59PM. To enhance the reproducibility of your results, we recommend that if applicable you deposit your laboratory protocols in protocols.io, where a protocol can be assigned its own identifier (DOI) such that it can be cited independently in the future. For instructions see: http://journals.plos.org/plosone/s/submission-guidelines#loc-laboratory-protocols

We look forward to receiving your revised manuscript.

Kind regards,

Giacomo Pucci

Academic Editor

PLOS ONE

Journal Requirements:

2. In your Methods section, please provide additional information about the participant recruitment method and the demographic details of your participants. Please ensure you have provided sufficient details to replicate the analyses such as: a) the recruitment date range (month and year), b) a description of any inclusion/exclusion criteria that were applied to participant recruitment, c)  a statement as to whether your sample can be considered representative of a larger population, d) a description of how participants were recruited, and e) descriptions of where participants were recruited and where the research took place. Moreover, please specify how the levels of physical activity were assessed.

Reviewers' comments:

Reviewer's Responses to Questions

**Comments to the Author**

1. Is the manuscript technically sound, and do the data support the conclusions?

Reviewer #1: Partly

Reviewer #2: Yes

2. Has the statistical analysis been performed appropriately and rigorously? 

Reviewer #1: Yes

Reviewer #2: Yes

3. Have the authors made all data underlying the findings in their manuscript fully available?

Reviewer #1: Yes

Reviewer #2: Yes

4. Is the manuscript presented in an intelligible fashion and written in standard English?

Reviewer #1: No

Reviewer #2: Yes

5. Review Comments to the Author

Reviewer #1: This is an interesting study on the possible differences between different methods of assessing central blood pressure levels in children, adolescents and young adults. Τhe authors aimed to analyze, in children, adolescents and young-adults (1) the agreement between cSBP, cPP, Pf and Pb obtained using carotid (CT) and radial tonometry (RT) and brachial-oscillometry (BOSC); and identify (2) explanatory factors for the differences between the approaches-data. 1685 subjects (mean/range age: 14/3-35 y.o.) assigned to three age-related groups (3-12; 12-18; 18-35 y.o.) were included. cSBP, cPP, Pf and Pb were assessed with BOSC (Mobil-O-Graph), CT and RT (SphygmoCor) records. Two calibration schemes were considered: MBPc and MBPosc for calibrations to similar BP levels. Correlation, Bland-Altman tests and multiple regression models were applied. Systematic and proportional errors were observed; errors´ statistical significance and values varied depending on the parameter analyzed, methods compared and group considered. The explanatory factors for the differences between data obtained from the different approaches varied depending on the methods compared. The highest cSBP and cPP were obtained from CT; the lowest from RT. The study is well-performed. There are several issues that if addressed could improve the manuscript.

Major comments

1. The length of the manuscript must be significantly reduced and especially the Discussion.

2. Please avoid having the figure legends and tables between the text because it gets really confusing trying to read through the manuscript. Please have a separate part with Tables and Figure legends.

3. In Table 1, for clarity and space reasons please remove the min and max columns as well as the interquartile range unless it is necessary due to a non-normal distribution. Moreover, include in the same column the mean and SD values. If this is done please present the overall and the other 3 smaller populations in a single table.

4. In Table 2, please include the mean and SD values in the same column. Please report the decimal values in the values in Table 2, because the p-values are different for similar differences and thus this means there is information missing. Moreover, when the value of p is ≥0.01, please report only 2 decimals.

5. Please place all the supplemental material in a different to the main manuscript file.

6. Table 3 needs to be reduced in size as well. I would keep only the r and p-values of the correlations and have the rest of the data in a supplemental table.

7. The authors reports errors as follows: “~2 to ~9”. The exact values should be included throughout the manuscript instead of using the ~ sign.

8. “Male sex showed the greatest differences between methods.” How can the authors explain this result?

9. “Differences in cSBP between RT and BOSC were explained by the BH of the subject.” How do the authors explain this result? Could be the fact that BOSC uses an algorithm that incorporates height for the assessment of cBPs?

10. The authors should report also their intra- and inter-observer reproducibility, because there is a chance that these errors in assessment between techniques are due to large variability in the assessment.

Minor comments

1. In the study population criteria please change “cardioactive” to “vasoactive”.

2. Please correct “have shown to met” to “have shown to meet”.

3. Please correct “("intantaneous blood pressure")” to “("instantaneous blood pressure")”.

4. Please correct “Pf y Pb” to “Pf and Pb”.

5. Please improve the resolution of all images because the cannot be properly assessed in their current format.

6. “Fig 4 (data included in S6 Table)”. Please report these data either on text or figure but not both.

7. Please change “were observed when using TR.” to “were observed when using RT.”.

8. Please report in the supplement the exact step by step method you used to conduct the wave separation analysis based on the observed aortic waveforms in each device and which given parameters you used from each device.

9. “When analyzing the age impact, the magnitude of the associations and differences varied depending on the approaches and calibration methods considered. For example, an increase in age equal to 10 years could be associated with a reduction in the differences in Pf equal to ~2 mmHg, ~4 mmHg, or ~6 mmHg, depending on whether RT-BOSC (cal. pDBP/MBPc), CT-BOSC (cal. pDBP/MBPc) or CT-BOSC (cal. pDBP/MBPosc) are compared.”

“A BH increase equal to 1 meter during growth would associate an increase in Pb differences equal to: ~2 mmHg for RT-CT (calibration: pDBP/MBPosc); ~7 mmHg and ~11 mmHg for RT-BOSC (calibration: pDBP/MBPc and pDBP/MBPosc, respectively), and ~5 and ~7 mmHg for CT-BOSC (calibration: pDBP/MBPc and pDBP/MBPosc, respectively).”

These parts seem more like a discussion rather than a result. If the authors want to keep it, they could move it in the Discussion section.

10. The authors in the beginning of their Discussion report more than 5 different results. They must choose which ones of them are the most essential and report only them, because it get really confusing with so many results in such a short space.

11. Please change “by Sharma et al.” to “by Sharman et al.”.

Reviewer #2: This is an interesting study exploring agreement between different methods (carotid and radial

tonometry and brachial oscillometry) in measuring central blood pressure and pulse wave analysis parameters in a large cohort of children, adolescents and young-adults.

The study is well conducted. The results highlighting the highest central BP levels obtained from carotid tonometry and the lowest by using radial tonometry are helpful when interpreting and comparing results of different studies.

I have two requests for the authors, which require further analysis which may improve the quality of the paper.

1) It seems that the way of calculating mean BP, necessary for calibration of all methods, may significantly change the estimation of central BP parameters. Is it possible to clarify which variables may influence the difference between MBPc and MBPosc, using a multiple regression model?

2) When interpreting the variables explaining absolute differences between methods (Table 4), a possibility is that actual BP levels (and particularly DBP) and heart rate may have a significant influence. My suggestion is to repeat the analysis in Table 2 by introducing these two parameters in the multiple regression models.

6. PLOS authors have the option to publish the peer review history of their article (what does this mean?). If published, this will include your full peer review and any attached files.

Reviewer #1: No

Reviewer #2: No

---

## [Author Response · Author response to Decision Letter 0]

7 Nov 2019

Academic Editor: Please ensure that your manuscript meets PLOS ONE's style requirements, including those for file naming.

Authors: We believe that the manuscript meets all the requirements. Thanks for the comments and suggestions that have contributed to improve the manuscript.

Academic Editor: In your Methods section, please provide additional information about the participant recruitment method and the demographic details of your participants. Please ensure you have provided sufficient details to replicate the analyses such as: a) the recruitment date range (month and year), b) a description of any inclusion/exclusion criteria that were applied to participant recruitment, c) a statement as to whether your sample can be considered representative of a larger population, d) a description of how participants were recruited, and e) descriptions of where participants were recruited and where the research took place. Moreover, please specify how the levels of physical activity were assessed.

Authors: As requested, in the revised version we provided additional methodological information.

Reviewer #1: This is an interesting study on the possible differences between different methods of assessing central blood pressure levels in children, adolescents and young adults. Τhe authors aimed to analyze, in children, adolescents and young-adults (1) the agreement between cSBP, cPP, Pf and Pb obtained using carotid (CT) and radial tonometry (RT) and brachial-oscillometry (BOSC); and identify (2) explanatory factors for the differences between the approaches-data. 1685 subjects (mean/range age: 14/3-35 y.o.) assigned to three age-related groups (3-12; 12-18; 18-35 y.o.) were included. cSBP, cPP, Pf and Pb were assessed with BOSC (Mobil-O-Graph), CT and RT (SphygmoCor) records. Two calibration schemes were considered: MBPc and MBPosc for calibrations to similar BP levels. Correlation, Bland-Altman tests and multiple regression models were applied. Systematic and proportional errors were observed; errors´ statistical significance and values varied depending on the parameter analyzed, methods compared and group considered. The explanatory factors for the differences between data obtained from the different approaches varied depending on the methods compared. The highest cSBP and cPP were obtained from CT; the lowest from RT. The study is well-performed. There are several issues that if addressed could improve the manuscript.

Authors: Thanks for your revision and comments.

Reviewer #1: The length of the manuscript must be significantly reduced and especially the Discussion.

Authors: Since we were asked (1) by the Editor to include more methodological information and (2) by Reviewer 2 to perform new analyses (which we did) it was difficult to reduce the length of the article.

However, as was requested, and without reducing the quality of the information provided, a significant part of the new analyses and the requested information (e.g. description of the wave separation analysis - WSA -) was included as Supplementary Material.

Reviewer #1: Please avoid having the figure legends and tables between the text because it gets really confusing trying to read through the manuscript. Please have a separate part with Tables and Figure legends. Please place all the supplemental material in a different to the main manuscript file.

Authors: We do understand the reviewer concern. However, that is what is indicated in the "Instructions for authors" (see below). We could have changed it, but the Editor has asked us to: "Please ensure that your manuscript meets PLOS ONE's style requirements, including .... ".

Supplementary Tables and Figures are in separate files. 

The only thing we have included at the end of the manuscript file (taking into account the "Instructions for authors") was the description of the supplementary files ("Supporting information captions").

Reviewer #1: In Table 1, for clarity and space reasons please remove the min and max columns as well as the interquartile range unless it is necessary due to a non-normal distribution. Moreover, include in the same column the mean and SD values. If this is done please present the overall and the other 3 smaller populations in a single table. In Table 2, please include the mean and SD values in the same column. Please report the decimal values in the values in Table 2, because the p-values are different for similar differences and thus this means there is information missing. Moreover, when the value of p is ≥0.01, please report only 2 decimals. Table 3 needs to be reduced in size as well. I would keep only the r and p-values of the correlations and have the rest of the data in a supplemental table.

Authors: As requested, in Table 1 we eliminated the minimum, maximum as well as the 25th and 75th percentile values. In addition, data was unified in a single Table. 

As suggested, Table 2 was modified (decimal values were added).

Table 3 was significantly shortened (its full version was included as a Supplementary Table). In the new (shortened) Table 3 we chose to keep the value of R and p of the correlations, and fundamental information of Bland-Altman tests, which allowed the existence of systematic (mean error, ME) and/or proportional errors to be assessed. In other words, "ME, C.I. 95% U.L. (mmHg)", "ME, C.I. 95% L.L. (mmHg)", "C.I. 95%, Upper limit (mmHg)" and "C.I. 95%, Lower limit (mmHg)" data (a total or 32 rows) were eliminated.

We kept mean and SD values in different columns due to we considered this enhances the visual clarity of the Table (mainly in the more extensive Tables - e.g. Table 2 -).

Reviewer #1: The authors reports errors as follows: “~2 to ~9”. The exact values should be included throughout the manuscript instead of using the ~ sign.

Authors: The requested change was done.

Reviewer #1: “Male sex showed the greatest differences between methods.” How can the authors explain this result? “Differences in cSBP between RT and BOSC were explained by the BH of the subject.” How do the authors explain this result? Could be the fact that BOSC uses an algorithm that incorporates height for the assessment of cBPs?

Authors: The issue was analyzed in greater detail in the revised version. Furthermore, the (additional) data analysis done in response to Reviewer #2 comments also contributed to explain the finding.

Reviewer #1: The authors should report also their intra- and inter-observer reproducibility, because there is a chance that these errors in assessment between techniques are due to large variability in the assessment.

Authors: The requested information was included summarily in the revised version. Detailed data was included in the Supplementary Materials´ section.

Reviewer #1: In the study population criteria please change “cardioactive” to “vasoactive”. Please correct “have shown to met” to “have shown to meet”. Please correct “("intantaneous blood pressure")” to “("instantaneous blood pressure")”. Please correct “Pf y Pb” to “Pf and Pb”. Please change “were observed when using TR.” to “were observed when using RT.”. Please change “by Sharma et al.” to “by Sharman et al.”.

Authors: The text was revised. Mistakes were corrected. Thanks.

Reviewer #1: Please improve the resolution of all images because the cannot be properly assessed in their current format.

Authors: As suggested, all figures were evaluated and corrected using the recommended system (Preflight Analysis and Conversion Engine (PACE) digital diagnostic tool, https://pacev2.apexcovantage.com/.)

Reviewer #1: 6. “Fig 4 (data included in S6 Table)”. Please report these data either on text or figure but not both.

Authors: We believe that both are useful, since Figure 4 is in the article (main text), highlighting most relevant haemodynamic information, while "Table S6" is a Supplementary Table included in Annexes (there, interested authors can find detailed numerical information). That is, Figure 4 does not show all the information included in Table S6. On the other hand, the inclusion of Table S6 in the manuscript (main text) would not make sense as the information contained is extensive and could be considered excessive in that context.

Reviewer #1: Please report in the supplement the exact step by step method you used to conduct the wave separation analysis based on the observed aortic waveforms in each device and which given parameters you used from each device.

Authors: As requested, a special document was included as Supplementary Material explaining the exact, step by step, method used to carry out wave separation analysis based on recorded carotid (SphygmoCor) and mathematically-derived aortic (SphygmoCor and Mobil-O-Graph) waveforms. We included examples of the analysis of records obtained in a real subject, using both softwares.

Reviewer #1: “When analyzing the age impact, the magnitude of the associations and differences varied depending on the approaches and calibration methods considered. For example, an increase in age equal to 10 years could be associated with a reduction in the differences in Pf equal to ~2 mmHg, ~4 mmHg, or ~6 mmHg, depending on whether RT-BOSC (cal. pDBP/MBPc), CT-BOSC (cal. pDBP/MBPc) or CT-BOSC (cal. pDBP/MBPosc) are compared.” “A BH increase equal to 1 meter during growth would associate an increase in Pb differences equal to: ~2 mmHg for RT-CT (calibration: pDBP/MBPosc); ~7 mmHg and ~11 mmHg for RT-BOSC (calibration: pDBP/MBPc and pDBP/MBPosc, respectively), and ~5 and ~7 mmHg for CT-BOSC (calibration: pDBP/MBPc and pDBP/MBPosc, respectively).” These parts seem more like a discussion rather than a result. If the authors want to keep it, they could move it in the Discussion section.

Authors: As suggested, the text was modified and included in the Discussion section.

Reviewer #1: The authors in the beginning of their Discussion report more than 5 different results. They must choose which ones of them are the most essential and report only them, because it get really confusing with so many results in such a short space.

Authors: As was requested, without reducing the quality of the information provided, the initial paragraph was modified (simplified) in order to clarify the message given.

Reviewer #2: This is an interesting study exploring agreement between different methods (carotid and radial tonometry and brachial oscillometry) in measuring central blood pressure and pulse wave analysis parameters in a large cohort of children, adolescents and young-adults. The study is well conducted. The results highlighting the highest central BP levels obtained from carotid tonometry and the lowest by using radial tonometry are helpful when interpreting and comparing results of different studies.

Authors: Thanks for the comments and suggestions that have contributed to improve the article.

Reviewer #2: I have two requests for the authors, which require further analysis which may improve the quality of the paper. 1) It seems that the way of calculating mean BP, necessary for calibration of all methods, may significantly change the estimation of central BP parameters. Is it possible to clarify which variables may influence the difference between MBPc and MBPosc, using a multiple regression model?

Authors: The requested analysis was performed in three steps. First, Bland-Altman analyses were performed to evaluate the agreement between MBPosc and MBPc (Supplementary Materials). Second, we identified demographic, anthropometric, cardiovascular risk factors and/or haemodynamic characteristics associated with the differences between MBPosc and MBPc (simple and point bi-serial correlation analysis) (new Table 5). Third, we included the selected independent variables (those with a p˂0.1) in multiple linear regression models (MLR) (new Table 6).

Reviewer #2: When interpreting the variables explaining absolute differences between methods (Table 4), a possibility is that actual BP levels (and particularly DBP) and heart rate may have a significant influence. My suggestion is to repeat the analysis in Table 4 by introducing these two parameters in the multiple regression models.

Authors: The suggestion was accepted. Thank you for your contribution. The requested analysis was performed and results were included in the new (modified) Table 4.

---

## [Decision Letter · Decision Letter 1]

26 Nov 2019

PONE-D-19-22199R1

Aortic pressure and forward and backward wave components in children, adolescents and young-adults: agreement between brachial oscillometry, radial and carotid tonometry data and analysis of factors associated with their differences

PLOS ONE

Dear Dr. Bia,

Thank you for submitting your manuscript to PLOS ONE. After careful consideration, we feel that it has merit but does not fully meet PLOS ONE’s publication criteria as it currently stands. Therefore, we invite you to submit a revised version of the manuscript that addresses the points raised during the review process.

Specifically, please shorten and edit the manuscript according to recommendations of Reviewer #1.

We would appreciate receiving your revised manuscript by Jan 10 2020 11:59PM. To enhance the reproducibility of your results, we recommend that if applicable you deposit your laboratory protocols in protocols.io, where a protocol can be assigned its own identifier (DOI) such that it can be cited independently in the future. For instructions see: http://journals.plos.org/plosone/s/submission-guidelines#loc-laboratory-protocols

We look forward to receiving your revised manuscript.

Kind regards,

Giacomo Pucci

Academic Editor

PLOS ONE

Reviewers' comments:

Reviewer's Responses to Questions

**Comments to the Author**

1. If the authors have adequately addressed your comments raised in a previous round of review and you feel that this manuscript is now acceptable for publication, you may indicate that here to bypass the “Comments to the Author” section, enter your conflict of interest statement in the “Confidential to Editor” section, and submit your "Accept" recommendation.

Reviewer #1: (No Response)

Reviewer #2: All comments have been addressed

2. Is the manuscript technically sound, and do the data support the conclusions?

Reviewer #1: Yes

Reviewer #2: Yes

3. Has the statistical analysis been performed appropriately and rigorously? 

Reviewer #1: Yes

Reviewer #2: Yes

4. Have the authors made all data underlying the findings in their manuscript fully available?

Reviewer #1: Yes

Reviewer #2: Yes

5. Is the manuscript presented in an intelligible fashion and written in standard English?

Reviewer #1: No

Reviewer #2: Yes

6. Review Comments to the Author

Reviewer #1: The authors have adequately answered most of the Editor’s and Reviewers’ comments. This paper has been improved. However, there are some small issues that still need their attention.

Comments

1. It is weird that the resolution in the Supplem. Figures is far better than the one in the main Figures. The labels are too blur and not visible for most of the Figures of the main manuscript.

2. The manuscript as commented in my first review is extremely long (reaching close to 10,000 words) and thus not easily readable. I was hoping that the authors would be able to shorten it after the revision, but despite their great efforts the length is not fit for publication. In fact, the number of supplemental tables (20) and figures (17) is huge. Despite, the good scientific background of the study the authors have not managed to report their results in a presentable manner. I think the authors should try and shorten the manuscript or if they think this is not feasible try to resubmit their findings in more than 1 articles.

3. One example of ways to reduce the size is the analysis presented in Table 6. There is no reason to use both enter and stepwise analysis in the models. Only the stepwise model would be sufficient.

Reviewer #2: The authors adequately answered to my previous comments.

The paper has improved from the previous version.

7. PLOS authors have the option to publish the peer review history of their article (what does this mean?). If published, this will include your full peer review and any attached files.

Reviewer #1: No

Reviewer #2: Yes: Andrea Grillo

---

## [Author Response · Author response to Decision Letter 1]

29 Nov 2019

29 -Nov-2019

Dear, Giacomo Pucci

Academic Editor

PLOS ONE

 I am sending the second revised version of the original manuscript (PONE-D-19-22199): "Aortic pressure and forward and backward wave components in children, adolescents and young-adults: agreement between brachial oscillometry, radial and carotid tonometry data and analysis of factors associated with their differences", to be considered for publication as full length article in Plos One.

 All reviewers’ suggestions were considered in the new version ('Response to Reviewers'; below). We thank the interest in our manuscript, and the reviewers’ comments that allowed us to increase the clarity of the manuscript.

 Sincerely,

Prof. Dr. Daniel Bia 

Physiology Department, School of Medicine. 

Universidad de la República, Montevideo, Uruguay

Phone: 5982-9243414 (3313); e-mail: dbia@fmed.edu.uy

Reviewer #1: The authors have adequately answered most of the Editor’s and Reviewers’ comments. This paper has been improved. However, there are some small issues that still need their attention.

Authors: Thanks for the comments and suggestions that contributed to improve the manuscript. 

Reviewer #1: 1. It is weird that the resolution in the Supplem. Figures is far better than the one in the main Figures. The labels are too blur and not visible for most of the Figures of the main manuscript.

Authors: As requested, in the revised version (1) Figures were improved (e.g. font size and graphics symbols were increased and the legends became more visible).

Reviewer #1: 2. The manuscript as commented in my first review is extremely long (reaching close to 10,000 words) and thus not easily readable. I was hoping that the authors would be able to shorten it after the revision, but despite their great efforts the length is not fit for publication. In fact, the number of supplemental tables (20) and figures (17) is huge. Despite, the good scientific background of the study the authors have not managed to report their results in a presentable manner. I think the authors should try and shorten the manuscript or if they think this is not feasible try to resubmit their findings in more than 1 articles. One example of ways to reduce the size is the analysis presented in Table 6. There is no reason to use both enter and stepwise analysis in the models. Only the stepwise model would be sufficient.

Authors: We understand the position of the Reviewer. In addition, we understand that while Plos One does not limit words, authors must be concise (*). In this context, we reduced the length of the article (approximately in 3,000 words). Then we consider it currently has a number of words that we believe is not excessive (Main text: Introduction + Methods + Results + Discussion + Conclusion: 5,136 words). See below the word count details.

It should be noted that we were asked for additional information and new (complementary) data analysis in the first revision of the original text. Consequently, the effort to shorten the work has been important.

We consider ed the example given by the reviewer, but eliminating ENTER method from Table 6 would only account for a reduction equal to 25-30 words. Taking into account this and that eliminating the analysis would reduce the quality of the manuscript and remove data already approved by Reviewer 2, we did not modify Table 6.

* Ploos One Instructions for authors: "Length: Manuscripts can be any length. There are no restrictions on word count, number of figures, or amount of supporting information.We encourage you to present and discuss your findings concisely."

 Word count

Section Previous revised version New (Actual) revised version

Abstract 250 250

Introduction 491 487

Material and Methods (*) 1561 1493

Results (*) 1266 1181

Discussion 1907 1783

Conclusion 195 192

SUBTOTAL (Introduction - Conclusion) 5420 5136

Acknowledgments 17 17

References 1214 (37 references) 1188 (37 references)

Supporting information 3512 805

TOTAL WORD FILE 10413 7396

* without considering tables and legend of figures

---

## [Editor Report · Decision Letter 2]

5 Dec 2019

Aortic pressure and forward and backward wave components in children, adolescents and young-adults: agreement between brachial oscillometry, radial and carotid tonometry data and analysis of factors associated with their differences

PONE-D-19-22199R2

Dear Dr. Bia,

We are pleased to inform you that your manuscript has been judged scientifically suitable for publication and will be formally accepted for publication once it complies with all outstanding technical requirements.

With kind regards,

Giacomo Pucci

Academic Editor

PLOS ONE
---

## [Editor Report · Acceptance letter]

11 Dec 2019

PONE-D-19-22199R2 

Aortic pressure and forward and backward wave components in children, adolescents and young-adults: agreement between brachial oscillometry, radial and carotid tonometry data and analysis of factors associated with their differences 

Dear Dr. Bia:

I am pleased to inform you that your manuscript has been deemed suitable for publication in PLOS ONE. Congratulations! Your manuscript is now with our production department. 

With kind regards,

on behalf of

Dr. Giacomo Pucci 

Academic Editor

PLOS ONE